# Resolving Lexical Bias in Model Editing

**Hammad Rizwan** [1]  **Domenic Rosati** [1]  **Ga Wu** [1]  **Hassan Sajjad** [1]

## Abstract

Model editing aims to modify the knowledge of a pre-trained language model. Previous approaches have often involved direct alterations to model weights, which can result in model degradation. Recent weight-preserving techniques avoid making modifications to the model's weights by employing an adapter that implements edits through auxiliary components. These rely heavily on scoping mechanisms based on distance functions on the model's representation space to determine when to trigger edits. We demonstrate that current adapter methods are *critically vulnerable* to strong lexical biases, leading to issues such as applying edits to irrelevant prompts with overlapping words. This paper presents a principled approach to learning a disentangled representation space that facilitates precise localization of edits by maintaining distance between irrelevant prompts while preserving proximity among paraphrases. In our empirical study, we show that our method, Projector Editor Networks for Model Editing - PENME, achieves state-of-the-art model editing results while being computationally efficient during inference compared to previous methods and adaptable across different architectures. We provide the codebase of PENME here: https://github.com/hammadrizwan/PENME.git

## 1. Introduction

Large Language Models (LLMs) are successful in solving a diverse range of natural language processing tasks (Devlin et al., 2019; Liu et al., 2019; Touvron et al., 2023b; Radford et al., 2019). Despite their successes, LLMs are fallible in large part due to the noisy and imperfect nature of the data used for training (Zhu et al., 2020). Moreover, as the world

[1]Department of Computer Science, Dalhousie University, Halifax, Canada. Correspondence to: Hammad Rizwan <hammad.rizwan@dal.ca>.

*Proceedings of the 42$^{nd}$ International Conference on Machine Learning*, Vancouver, Canada. PMLR 267, 2025. Copyright 2025 by the author(s).

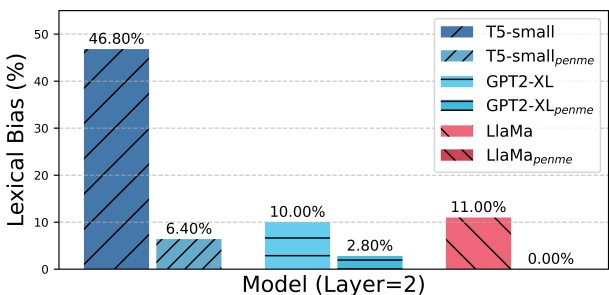

*Figure 1.* Projector networks mitigate lexical bias: a critical problem in adapter-based model editing techniques. Percentage of samples where irrelevant but lexically similar prompts **are closer** than semantically similar paraphrases in the representation space before and after our learned projection (PENME).

evolves, new information requires updates to the models, e.g. the leader of a country may change over time.

Periodically updating LLMs using fine-tuning is one potential solution, however, it risks degradation in performance, leading to training from scratch to preserve the model's original performance (Luo et al., 2023; Wang et al., 2025). However, retraining is often impractical due to the need of substantial computational resources, time, data, and labor.

Model editing has been proposed as a sample and compute efficient way to update LLMs (Yao et al., 2023). Historically *weight-modifying* techniques that make surgical small parameters updates such as ROME (Meng et al., 2022) have been popular. However, these techniques are known to be computationally inefficient (Yu et al., 2024) and result in catastrophic forgetting (Gupta et al., 2024a), the full impact of which is difficult to determine (Rosati et al., 2024).

***Weight-preserving*** approaches solve these problems by maintaining the original model parameters and use additional components like key-value codebook adapters (GRACE and MELO) (Hartvigsen et al., 2023; Yu et al., 2024) that apply a scoping mechanism to determine whether to trigger a model edit, using a distance function over model representations to assess semantic similarity between the input and an edits registered in a codebook.

We find that the performance of weight-preserving methods is heavily reliant on *scoping mechanism* which suffers from a critical vulnerability of **Lexical bias** (Figures 1 and

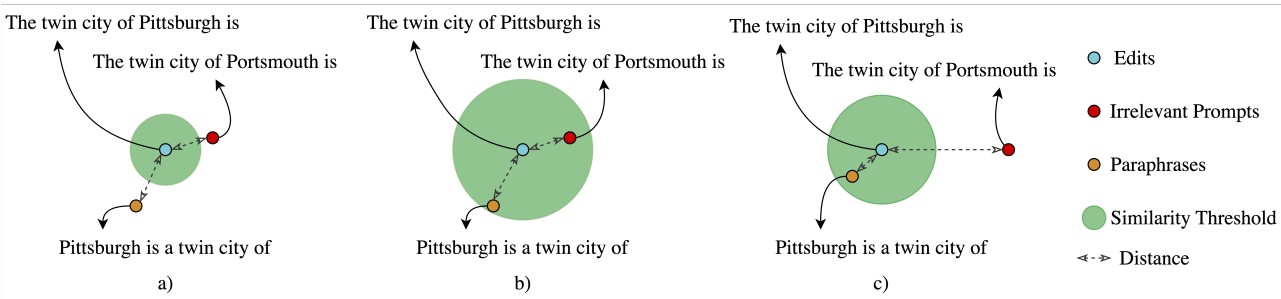

*Figure 2.* An illustration of lexical bias in embeddings: a) a low similarity threshold (illustrated with the circle) results in failing to edit paraphrases. b) A high similarity threshold results in misfires with irrelevant prompts. c) illustrates our solution which restructures the representation space.

2), *prompts with similar lexical tokens but different semantics that are closer together in the representation space compared to a prompt and its respective paraphrases*. Lexical bias prevents current adapter-based methods from effectively being able to balance generalization to unseen paraphrases and "misfiring" on semantically dissimilar (irrelevant) prompts. Our analysis (§ 6.1; Figure 4) of these misfires reveals that the representation space used to calculate semantic similarity is dominated by a lexical bias where irrelevant prompts (e.g. "The twin city of Portsmouth is") are often closer by euclidean distance to an input (e.g. "The twin city of Pittsburgh is") than a true semantically similar prompt (e.g. "Pittsburgh is a twin city of").

Based on this analysis, we propose Projector Editor Networks for Model Editing (PENME), an advancement over adapter-based model editing that explicitly targets the lexical bias problem by learning a projection that disentangles lexically similar and semantically similar text representations. We empirically demonstrate that resolving lexical bias enables high edit generalization performance across paraphrases of a prompt while also achieving strong locality that prevents irrelevant prompts from triggering edits.

Our contributions are as follows: **(1)** We show that representations extracted across layers exhibit lexical bias, showing a bias towards token overlap, which introduces significant challenges for adapter-based model editing techniques. **(2)** We propose PENME, a model editing framework that learns a projection network that maps the model's representation space to a new representation space where lexical bias is minimized. **(3)** We integrate our projection network in an adapter-based retrieval scheme for model editing, demonstrating, for the first time in adapter-based approaches, high efficacy in both paraphrase execution (generalization) and prevention of misfires on irrelevant prompts (locality).

## 2. Related Work

There are a number of surveys that provide a comprehensive overview of various model editing methods (Wang et al.,

2024a;c; Mazzia et al., 2024; Durrani et al., 2025). Here, we summarize three main themes of model editing.

**Weight-modifying** approaches apply targeted updates to the model's weights to reflect new knowledge (Tanno et al., 2022; Meng et al., 2022; Gupta et al., 2023; Xu et al., 2023; Hase et al., 2023b; Li et al., 2024b; Fang et al., 2025; Ma et al., 2025). Fine-tuning on edited knowledge can have a detrimental impact on a model's general capabilities; therefore, many methods instead perform targeted updates. These approaches typically rely on the localization hypothesis (Miller et al., 2016; Geva et al., 2021) in the Transformer architecture, which conjectures that the point-wise feed-forward components act as a key–value memory for information retention within an LLM, a claim that has recently been challenged, in the context of model editing by Hase et al., 2023a. Meng et al. (2022; 2023, ROME, MEMIT) identifies salient neurons within the feed-forward layers, facilitating targeted updates to effect the desired edits using causal analysis. Similarly, Li et al. (2024b, PMET) investigates the role of multi-headed attention, in conjunction with feed-forward layers, for model editing. As we mentioned earlier, these methods suffer from general model degradation due to gradual performance drift, which can lead to catastrophic forgetting (Gupta et al., 2024a). Improvements on MEMIT have been made by Ma et al. (2025, PRUNE), which bounds the condition number of the edited weight slice so that successive edits cause only minimal drift and the model's overall behaviour stays intact. Similarly, Fang et al. (2025) propose AlphaEdit, which projects each update into the null space of knowledge that must remain unchanged. This preserves existing knowledge while inserting the update. AlphaEdit showcases the strongest performance; however, the method is evaluated on mini-batches of 100 edits, reducing the batch size to 1 for sequential editing, this necessitates recomputing the projection for every edit, increasing computational cost and is likely to lead to performance drift during prolonged editing.

An alternative approach, using a hypernetwork, is Mitchell et al. (2021, MEND), which predicts new model weights by

generating low-rank decompositions of the weight matrices across different layers.

**Weight-preserving pre-input** approaches depend on extracting and processing relevant edit information before the input is processed by the main model Mitchell et al. (2022); Zheng et al. (2023); Zhong et al. (2023). For example, SERAC (Mitchell et al., 2022) employs a memory-based model editing strategy augmenting with a memory storage and supplementary models to determine the scope of the edit. RAG-based methods like IKE (Zheng et al., 2023) leverage similarity-based retrieval to extract and rank edit demonstrations from memory and use in-context reasoning to edit. A limitation of these approaches is the computational overhead as they require additional models for ranking, relevancy, context processing and generalization.

**Weight-preserving post-input** These rely on the model's internal representations to implement *scoping mechanisms*, determine whether a specific edit applies for the current input. If an edit does apply, they employ a playback mechanism such as representation vector addition or replacement that results in the model generating updated outputs Hartvigsen et al. (2023); Yu et al. (2024); Lee et al. (2022). Most methods employ a key-value codebook: a vector representation of the inputs which should be edited are stored as a key and the representation vectors of the desired edit phrase are stored as a value. Semantic similarity is computed using a distance metric, such as Euclidean distance, against future inputs to the language model. If a distance threshold is satisfied, then the edit vector is "played-back."

Alternative strategies introduce lightweight auxiliary structures—either additional neurons to scope and steer model outputs, as in Huang et al. (2023); Zhu et al. (2024), fixed-size hook layers that accumulate residual updates for consecutive batch edits while keeping the base weights frozen Li et al. (2024a), purpose-built external memories Wang et al. (2024b, WISE), or compact jet-pack modules (Sutton et al., 2024) that modify hidden activations for a specific prompt, originally proposed for stealth edits.

**Lexical Bias** Hartvigsen et al. (2023, GRACE) employs playback vectors as above, whereas Yu et al. (2024, MELO) utilizes LoRA blocks. We find that both of these methods are vulnerable to lexical bias which invalidates their retrieval methods. In order to resolve this issue, current methods need to hand-engineer distance thresholds which balance the trade off generalization for irrelevant prompt misfiring protection. Our method, PENME, resolves lexical bias by learning a new disentangled representation space where large thresholds can safely be used that are much more reliable at preventing misfires. While lexical bias has been repeatedly documented through initial diagnoses of syntactic and lexical heuristics (McCoy et al., 2019; Dumpala et al., 2024), dataset-level analyses (Zhou & Bansal, 2020),

and adversarial stress tests (Nie et al., 2020), recent work shows the bias remains stubborn (Serrano et al., 2023). Notably, our work is the first to highlight this issue in model editing and propose a solution to mitigate it.

## 3. Problem Setting: Model Editing

The aim of model editing is to alleviate the need for complete retraining when updating learned knowledge. Editing attempts to satisfy the following conditions: (1) sample efficiency: update the model with the fewest number of samples possible, (2) compute efficiency: train a small portion of the model only, (3) minimal impact: make as small of an impact on unrelated behaviour as possible i.e. prevent misfires on irrelevant prompts and (4) ensure generalization: maintain accurate paraphrase behaviour i.e. retrieval of correct edits in adapter-based approaches.

The goal is to modify the behaviour of a model $M$ on a dataset $D = [d_1, ..., d_n]$ where the sample $d_i$ is the tuple $(x_i, y_i, [p_{i1}, ..., p_{in}], [p_{i1}^{\neg}, ..., p_{in}^{\neg}])$, $x_i$ is the edit prompt, $y_i$ is the new output tokens, $p_{i,1:n}$ are a set of paraphrases of the edit prompt $x_i$ and $p_{i,1:n}^{\neg}$ are *irrelevant prompts*, examples that are both lexically and semantically related; however, they represent cases where the underlying model's generation output should remain unchanged. For instance, consider the edit "What is the twin city of Detroit". A lexically similar prompt would be "What is the twin city of London", whereas a semantically related prompt might be "For Detroit, tell me what twin city it has" where the semantic relationship lies in the fact it is a paraphrase. For successful model editing, the edited model, $M'$, should generate new target tokens $y_i$ for a specific input $x_i$ (Edit Success) and its related paraphrases $p_{1:n}$ (Generalization), while maintaining the model's behaviour on semantically unrelated prompts $p_{1:n}^{\neg}$ (Locality). The following metrics illustrate how these factors are typically operationalized (see for example Yao et al., 2023; Yu et al., 2024; Hartvigsen et al., 2023; Gupta et al., 2024b).

**Edit Success (ES):** The proportions of edits that the model is able to recall or generate correctly, also referred as efficacy, reliability, and edit score. Formally we say $M'(x_i) = y_i, \forall(x_i, y_i) \in d_{1:n}$.

**Locality:** The proportion of irrelevant prompts for which the model generates the same outputs prior to editing, also referred to as specificity, neighbourhood success, retain rate and neighbourhood score and is denoted as:
$M'(p_{ij}^{\neg}) = M(p_{ij}^{\neg}), \forall p_i^{\neg} j \in d_{1:n}$.

**Generalization:** The proportion of paraphrases for which the model is able to recall or generate the correct edited information, also described as paraphrase success: $M'(p_{ij}) = y_i, \forall p_j \in p_i, \forall(p_i, y_i) \in d_{1:n}$.

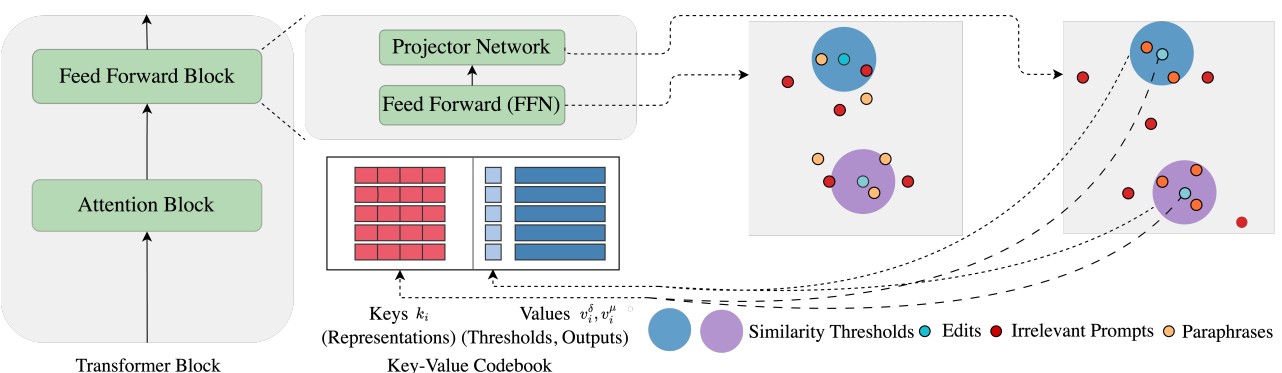

*Figure 3.* **PENME** uses a projection network that interfaces with the pointwise feed-forward layer output in a transformer block. The projection network, coupled with key-value codebook storage, acts as a scoping mechanism by comparing projection outputs with codebook entries. This determines whether the current input relates to a specific edit or should pass through the model unmodified.

**Score:** It is the mean of the above three metrics and is used for benchmarking.

## 4. Projector Editor Networks for Model Editing (PENME)

PENME is a weight-preserving model editor that leverages an adapter module which is integrated after the pointwise feed-forward layers within a transformer block of a pre-trained LLM. By introducing this additional component rather than altering the original model weights, PENME enables the integration of new information while preserving the LLM's initial capabilities.

PENME, illustrated in Figure 3, consists of two components: (1) **Projection Network** ($g$) projects model activations denoted $h_l(input)$ at layer $l$ into a distinct representation space $g(h_l(input))$. (2) **Key-Value Codebook** stores the projected model activations $g(h_l(input))$ at layer $l$ as keys and corresponding values containing a learned similarity threshold ($\delta$) and the new associated output information $y_i$. This paper only considers storing strings as $y_i$, but vectors (Hartvigsen et al., 2023) or LoRA block indices (Yu et al., 2024) can also be stored as values, which facilitate playback approaches.

In the following sections, the vectors $\vec{x_{ij}}$, $\vec{p_{ij}}$, and $\vec{p_{ij}}$ are outputs of the projection network $g(h_l(input))$ for dataset components $x_i$, $p_{ij}$ and $\vec{p_{ij}}$.

### 4.1. Projection Network

We hypothesize that if the representation space suffers from lexical bias, then we could learn a new representation space that *disentangles* lexical and semantic representations. We achieve this by training a projection network $g(\cdot) : \mathbb{R}^d \to \mathbb{R}^d$ using contrastive learning whose function is to project inputs into a space where paraphrases of inputs are closer to

edited inputs than irrelevant prompts. Our training loss is inspired by contrastive learning (Hadsell et al., 2006) and is defined by the following loss function:

$$\mathcal{L}(\vec{x_i}, \vec{z}) = (1-t)\frac{1}{2}||\vec{x_i} - \vec{z}||_2^2$$
$$+ t\frac{1}{2}\left[\max(0, m - ||\vec{x_i} - \vec{z}||_2)\right]^2, \quad (1)$$
$$t = \begin{cases} 1, & \text{if } \vec{z} \leftarrow \vec{p_{ij}}, \\ 0, & \text{if } \vec{z} \leftarrow \vec{p_{ij}} \vee \vec{x_l}. \end{cases}$$

where $t$ is the target $\{0, 1\}$ which is 0 when the training pair is $\{x_i, p_{ij}\}$ (edit, paraphrase) and 1 when the training pair is $\{x_i, \vec{p_{ij}}\}$ (edit, irrelevant) or the inter-edit (or edit-to-edit) pair $\{x_i, x_l\}$ where we sample an unrelated edit, $m$ is the margin which pushes $\vec{p_{ij}}$ at least $m$ distance away from $\vec{x_i}$. The projection network is trained such that for all samples in a dataset, edits $x_i$ and edit paraphrases $p_{ij}$ are close together while edits $x_i$ and irrelevant $\vec{p_{ij}}$ paraphrases or unrelated edits $x_l$ are pushed apart in the projection space. Training is performed by sampling pairs at random. Note that $\vec{z}$ is a variable that is assigned either a paraphrase, an irrelevant prompt, or an unrelated edit just as a way to make the loss function more concise.

The inherent lexical and semantic similarities among edits increase the probability of certain edit paraphrases exhibiting greater proximity to other unrelated edits. This phenomenon can lead to erroneous paraphrase-edit associations during execution, potentially triggering inappropriate edit operations. This is why we also push unrelated edits farther away in Eq. 1 as well as unrelated prompts. These pairings are formed based on a similarity threshold defined as a hyperparameter $\phi$.

The projector network is a 2-layer MLP with one ReLU non-linearity and batch norm applied between each layer. The dimensionality of each layer is the same as the original

representation space. Note that this network is only applied to one single layer. The compact architecture of the projection network enables it to be trained on GPUs with limited memory capacity since we can amortize the computation of representation vectors (denoted above with the $\vec{vec}$ symbol beforehand) irrespective of the underlying model's scale. We provide the details of implementation, data construction and training in Appendix A.

### 4.2. Key-Value Codebook

The key-value codebook is a memory mechanism designed to store edits and their corresponding outputs. For each edit, representations are generated by passing the input $x_i$ through the model and the projection network, denoted as $\vec{x_i} = g(h_l(x_i))$. $\vec{x_i}$ are then stored as keys $k_i \in K$ in the codebook and are utilized during runtime in a similarity-based retrieval system to access the relevant edit. The codebook value $v_i \in V$ consists of the edited information $\mu$ along with a similarity threshold $\delta$. The edited information in this paper is an exact string stored from the model editing dataset. The threshold serves as a scoping mechanism and is learned using a procedure described below. For a given input prompt $x_i$, euclidean distance $|| \cdot ||_2$ is computed with all keys in the codebook. From the computed distances, we determine if the input prompt $x_i$ is relevant to the edited codebook value $v_i^\mu$ and its corresponding threshold $v_i^\delta$. This is expressed as:

$$\underset{k_i \in K}{\arg\min} \quad \|\vec{x_i} - k_i\|_2$$
$$\text{s.t.} \quad \|\vec{x_i} - k_i\|_2 < v_i^\delta \tag{2}$$

If the prompt $x_i$ is deemed relevant (Equation 2), the output information of the edit is retrieved from codebook $v_i^\mu$. Otherwise, the typical model output $M(x_i)$ is employed.

### 4.3. Finding the thresholds $v_i^\delta$ and $\tau$

Initial experimental findings regarding the thresholds $v_i^\delta$ reveal that unseen test paraphrases typically demonstrate greater distance than the average seen training paraphrases, while the inter-paraphrase distances within the training set exhibit variation across edits. In contrast, unseen test irrelevant prompts generally show closer proximity to edits compared to the nearest seen training irrelevant prompts. This effect is illustrated in greater detail in Appendix B. We determine an appropriate threshold by utilizing a data-driven thresholding scheme based on the training data:

$$v_i^\delta = \text{Max}\left(\|\vec{x_i} - \vec{p_{ij}}\|_2\right) + \tau \tag{3}$$

The threshold is determined as the maximum paraphrase distance observed for each individual edit, augmented by a hyperparameter $\tau$ to account for unseen paraphrases. We select a value of $\tau$ through grid search. This formulation allows our method to achieve an optimal balance between generalization and locality preservation. Alternatively, another

possibility is to set the threshold based on close irrelevant prompts $Min(\|\vec{x} - \vec{p_{ij}}\|_2) - \tau$; this option would maintain locality by preserving all training irrelevant prompts.

### 4.4. Analysis of Codebook Management and Scalability

Both GRACE and MELO require multiple paraphrases added to the codebook to improve generalization. The PENME codebook scales linearly with the number of edits, as each edit corresponds to a single codebook entry. Maintaining one entry per edit enables efficient edit removal or updates, providing greater flexibility in edit management. In contrast, the scoping mechanism employed by Hartvigsen et al. (2023); Yu et al. (2024) to deal with multiple, possibly conflicting, entries per codebook requires splitting and merging operations. The effectiveness of this approach varies across datasets. For instance, for GRACE, the zsRE dataset exhibits a high occurrence of similar edit outputs (same entity with the same edit), allowing for substantial reductions in codebook entries. Specifically, 1,000 edits on zsRE require only 658 entries, whereas the Counterfact dataset requires 1,682 entries for just 300 edits. The combination of this consolidation process and the potential for edits to be closely related in vector space leads to overlapping cluster radii, necessitating cluster size reduction. This inadvertently results in the removal of certain edits. A detailed comparison between PENME and the scoping mechanisms employed by GRACE and MELO is presented in Appendix C. The results demonstrate that PENME achieves superior edit retrieval speed and highlights the problem of edit conflict and edit forgetting by GRACE and MELO.

## 5. Experimental Setup

We assess the performance of PENME across a spectrum of transformer-based LLMs, including Text-to-Text Transfer Transformer (specifically T5-small) (Raffel et al., 2020), Llama-2-7b (Touvron et al., 2023a) and GPT2-XL (Radford et al., 2019). We compare PENME with GRACE and MELO, as these are the only other current weight-preserving adapter-based methods. Additionally, we include MEMIT and SERAC[1]. Working details of the methods and hyper-parameters are provided in Appendix D.1. We utilize the methodology outlined in §6.1 to select an optimal layer to introduce PENME adapter and use the second layer for all models. We determine the optimal threshold for each edit by systematically varying the $\tau$ parameter in Eq. (3) across a range of 0.05 to 0.20.

**Dataset** The zsRE dataset (Levy et al., 2017) and the Counterfact dataset (Meng et al., 2022) are the most commonly used model editing datasets. zsRE consists of an

---

[1]A simpler version of SERAC is used in Hartvigsen et al. (2023) called Defer.

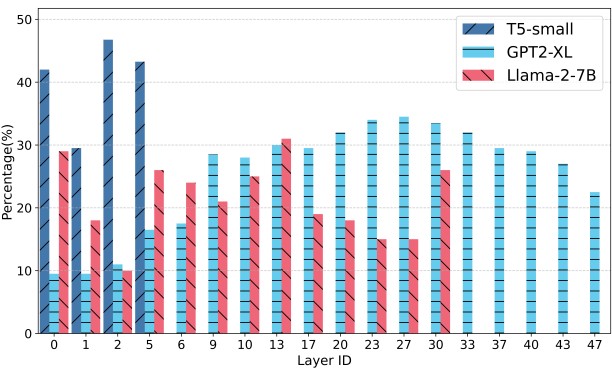

*Figure 4.* Percentage of samples *where edits are closer* to lexically similar yet irrelevant prompts as compared to paraphrases in the representations space of different models across various layers. T5-small, GPT2-XL and Llama-2-7b have 6, 32, and 48 layers, respectively. The full figure for all layers is presented in App. E.1.

edit prompt along with several paraphrased versions of that prompt. Irrelevant prompts are sourced from the NQ dataset (Kwiatkowski et al., 2019), which offers a wide range of user query questions. In contrast, Counterfact has similar edit and paraphrase prompts but employs a more nuanced approach to irrelevant prompts. It includes prompts that are similar to the edit prompt in both semantic nature and lexical structure. This differs significantly from zsRE, where the irrelevant prompts are neither semantically nor lexically related to the edit prompt. Moreover, zsRE has a lower diversity in subjects, relationships, and linguistic variations (Meng et al., 2022). This structural difference between the datasets has important implications for evaluation. In zsRE, the lack of semantic or lexical relationships between the edit prompt and its irrelevant prompts allows weight-preserving approaches to achieve high locality scores with relative ease. The enhanced complexity of Counterfact renders it a more robust benchmark for evaluating editing mechanisms. Dataset processing and training data construction details are provided in Appendix D.2.

**Downstream Tasks** We adopt the evaluation setup of Ma et al. (2025) to assess downstream performance. Specifically, we evaluate using three tasks: sentiment classification using the DAiR-Emotions dataset (Saravia et al., 2018), summarization using the CNN/DailyMail dataset (Hermann et al., 2015), and natural language inference (NLI) using the RTE dataset (Dagan et al., 2005).

## 6. Evaluation

This section presents the evidence of lexical bias, the results of PENME in achieving separability of irrelevant prompts and paraphrases, and a comparison with other methods.

### 6.1. Lexical Bias

To examine the lexical bias of representations, we randomly sampled 500 entries from the Counterfact dataset (see §5). For each entry, we created triplets consisting of an edit prompt, a randomly sampled paraphrase prompt and an irrelevant prompt with ***high lexical overlap*** $(x_i, p_i, p_i^-)$. In Table 7, we see the ROUGE-1 token overlap w.r.t $x_i$ mean F1 was 0.3 for both $p_i$ and $p_i^-$. Qualitative samples are provided for the reader's validation in Table 8. These triplets are fed into various models, and representation vectors $(\vec{x_i}, \vec{p_i}, \vec{p_i^-})$ from the feed-forward block of each layer $l$ are extracted. We select either averaged token representations or dedicated sentence representations, based on whether a given model offers a specific token for sentence-level representation. We calculate two sets of pairwise Euclidean distances: (1) Between edit representations and paraphrase representations: $||\vec{x_i} - \vec{p_i}||_2$ (2) Between edit representations and irrelevant prompts representations: $||\vec{x_i} - \vec{p_i^-}||_2$. We then compare these distances to determine if irrelevant prompts are closer to the edits than the paraphrases $||\vec{x_i} - \vec{p_i}||_2 > ||\vec{x_i} - \vec{p_i^-}||_2$. Figure 4 displays the percentage of samples where irrelevant prompts *were closer* to the edits.

Figure 4 reveals an intriguing pattern: except for the first layer in most models, the early layers demonstrate a reduced percentage of samples where irrelevant prompts are closer to edits than paraphrases. However, the trend shifts as we progress through the model's depth. In the mid-layers, this percentage begins to ascend once more, only to descend slightly towards the final layers, albeit with subtle fluctuations among them. We hypothesize that in the initial layers, token-specific information remains largely isolated. However, as the input traverses deeper into the model, guided by repeated attention mechanisms, this information becomes amalgamated across tokens (Sajjad et al., 2022). Moreover, repeated normalization as demonstrated by Takase et al. (2022) results in smaller changes in weights of an LLM, leading to embedding vectors in the final layers being similar, thus only subtle fluctuations are seen in the percentages.

These results indicate why there is a significant chance of misfire in adapter-based methods: lexical bias. This also provides a systematic approach for identifying the optimal layer to introduce PENME integration by elucidating the regions within the model's architecture where lexical bias exhibits minimal influence. Although the projector network approach can be generalized across all layers, as demonstrated in Appendix E.2, it is advantageous in terms of training time to integrate at points of minimal influence.

### 6.2. Disentangled Projection Space

In this section, we validate our proposed projection network in its ability to learn a generalized disentangled representation space where paraphrases are closer to edits as compared

*Table 1.* A comparative analysis of PENME and recent model editing methods on 2000 edits from the Counterfactual dataset and 1000 edits on zsRE. The metrics are Edit Success (ES), Locality (Loc) and Paraphrase Generalization (Para).

| Method | Model | COUNTERFACT | | | | zsRE | | | |
|---|---|---|---|---|---|---|---|---|---|
| | | ES | Loc | Para | Score | ES | Loc | Para | Score |
| PENME | T5-small | **1.000** | 0.787 | **0.808** | **0.865** | **1.000** | **0.941** | 0.913 | **0.951** |
| | Llama-2-7b | **1.000** | 0.869 | **0.906** | **0.925** | **1.000** | **0.987** | **0.966** | **0.984** |
| | GPT2-XL | **1.000** | 0.847 | **0.875** | **0.907** | **1.000** | **0.957** | 0.940 | **0.966** |
| MELO | T5-small | 0.850 | 0.800 | 0.037 | 0.562 | 0.990 | 0.640 | **0.986** | 0.872 |
| | GPT2-XL | **1.000** | **1.000** | 0.020 | 0.673 | **1.000** | 0.004 | **1.000** | 0.668 |
| GRACE | T5-small | **1.000** | **0.860** | 0.140 | 0.667 | **1.000** | 0.730 | **0.993** | 0.907 |
| | Llama-2-7b | **1.000** | **0.997** | 0.002 | 0.666 | 0.100 | 0.591 | 0.000 | 0.230 |
| | GPT2-XL | **1.000** | 0.996 | 0.003 | 0.666 | 0.992 | 1.000 | 0.010 | 0.667 |
| SERAC | T5-small | 0.017 | 0.526 | 0.010 | 0.184 | 0.017 | 0.526 | 0.010 | 0.184 |
| | Llama-2-7b | 0.992 | 0.372 | 0.651 | 0.672 | **1.000** | 0.114 | 0.357 | 0.490 |
| | GPT2-XL | 0.947 | 0.669 | 0.408 | 0.675 | 0.474 | 0.003 | 0.811 | 0.429 |
| MEMIT | Llama-2-7b | 0.147 | 0.149 | **1.000** | 0.432 | 0.402 | 0.002 | **1.000** | 0.468 |
| | GPT2-XL | 0.785 | 0.788 | 0.502 | 0.692 | 0.214 | 0.000 | **1.000** | 0.405 |
| FT | T5-small | 0.955 | 0.000 | 0.450 | 0.468 | 0.017 | 0.526 | 0.010 | 0.184 |
| | Llama-2-7b | 0.404 | 0.393 | 0.417 | 0.405 | 0.569 | 0.020 | 0.746 | 0.445 |
| | GPT2-XL | 0.968 | 0.851 | 0.395 | 0.738 | 0.608 | 0.005 | 0.889 | 0.501 |
| PENME$_{stream}$ | T5-small | 1.000 | 0.782 | 0.756 | 0.846 | 1.000 | 0.615 | 0.550 | 0.721 |
| | Llama-2-7b | 1.000 | 0.871 | 0.818 | 0.896 | 1.000 | 0.716 | 0.792 | 0.836 |
| | GPT2-XL | 1.000 | 0.850 | 0.768 | 0.872 | 1.000 | 0.733 | 0.768 | 0.833 |

to irrelevant prompts. We sample 1500 tuples $(e_i, p_i, p_i^{\neg})$ of edits denoted $e_i$, paraphrases $p_i$, and their unrelated irrelevant prompts $p_i^{\neg}$ from the Counterfact dataset with accompanying input prompts $x_i$ and split them into train and test sets of 1000 and 500 samples respectively. We use the training set to train the projector network using model representations from layer 2 of each model. To evaluate the network's performance, we compare two types of test representations: the original model representations $h_l(x_i)$ where $x_i$ is the input prompt and the projected representations $g(h_l(x_i))$. This comparison uses the experimental method described earlier, allowing us to determine whether the projection network successfully learns to create a lexically disentangled representation space.

The results presented in Figure 1 demonstrate that the projector network effectively learns to distance lexically similar but unrelated irrelevant prompts in comparison to paraphrases. A two-dimensional PCA visualization of the representation space, illustrating this phenomenon, is provided in Appendix F.2.

For data pairs where irrelevant prompts are closer to edits than paraphrases, T5-small exhibits a dramatic decrease from $46\%$ to $6.4\%$. Similarly, GPT2-XL reduces from $10\%$ to $2.8\%$, and Llama-2-7b drops to $0\%$ from $11\%$, indicating perfect separability of irrelevant prompts and paraphrases.

### 6.3. Model Editing Results

Table 1 presents the comparative results of PENME and recent model editing methods for 2000 edits on the Counterfact dataset and 1000 edits on zsRE.[2] PENME demonstrates stable performance across editing metrics as compared to other model editing approaches. In particular, PENME shows high efficacy on both locality and generalization and has stable performance across different models. Observe that for both GRACE and MELO, these methods require trading of locality for paraphrase performance or vice versa due to lexical bias.

GRACE, similar to PENME, demonstrates high edit success rates due to its inherent design. However, its generalization scores compared to PENME were markedly low, suggesting poor performance on edit paraphrases post-editing. GRACE achieved the highest locality scores, with T5-small at 0.92 and Llama-2-7b nearly perfect at 0.997. The substantial difference between locality and generalization scores can be attributed to GRACE's use of a very low distance threshold, resulting in poor performance on paraphrases but successfully avoiding irrelevant prompts spillover into edits.

SERAC also achieves a high edit success but shows mixed performance results for generalization and locality across models. For T5-small, the approach does not work well as

---

[2]Due to computational constraints, editing is performed on 1,000 zsRE samples. However, this number is consistent with the typical sample size used in related literature.

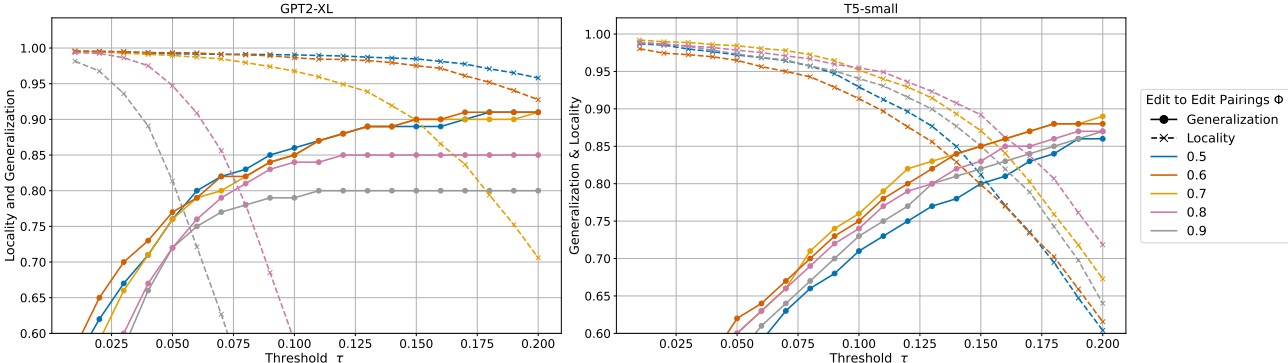

*Figure 5.* Shows the trade-off between generalization and locality performance across different hyperparameter settings. The distance threshold $\tau$ varies from 0.01 to 0.2 (0.01 increments and $\tau$ is normalized by 100), while the edit-pairing similarity threshold $\phi$ ranges from 0.5 to 0.9 (0.1 increments). Higher $\phi$ values enforce stricter edit similarity requirements. The results showcase the effect of hyperparameter tuning on the projector network's learning capacity and overall performance.

SERAC uses logically entailed facts to determine the scope, the original work uses a T5-large which is significantly better at reasoning.

For GPT2-XL, MEMIT demonstrates moderate effectiveness, achieving an edit success rate of 0.785 and a locality score of 0.788. In contrast, when applied to Llama-2-7b, both the edit success and paraphrase success rates are relatively low, although the locality score remains high. This discrepancy is likely due to challenges stemming from MEMIT's training on the Llama-2-7b model, as similar findings have been reported by Huang et al. (2024).

**Projector Generalization** In the previous setting, we trained the projector using all of the edit samples at once i.e. batch editing. In this setting, we evaluate the stream or lifelong editing setting for zero-shot generalization, where we update the codebook once per edit using a frozen projector. As a trained projection network is needed we initialize PENME$_{stream}$ using 2k unseen samples from Counterfact. Table 1 presents the results. Analysis on the zsRE dataset demonstrates the projector network's capacity for zero-shot generalization, achieving robust performance metrics while maintaining equilibrium between generalization and locality. For the Counterfact dataset, the drop in performance is minimal with the slight exception on the generalization of GPT2-XL. These findings suggest that training a projector network on a more extensive dataset to reduce lexical bias could enable its use as a modular component.

### 6.4. Scaling Edits

We evaluate the projection network's stability under varying numbers of edits using incrementally larger training sets ranging from 1000 to 5000 edits, with 1000-edit increments per training session. The results of the experiment are shown in Figure 6. Projector network trained on representations from T5-small demonstrates lower overall performance in

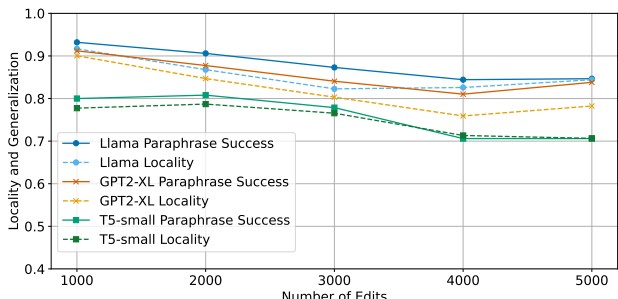

*Figure 6.* PENME's performance in terms of Locality (dotted) and Generalization (continuous line) across varying numbers of edits

generalization and locality compared to other models. We hypothesize that this under-performance may be attributed to either the model's smaller size, resulting in less robust learned representations, or the fact that it was trained on a more limited dataset relative to larger, more recent models. Projection networks trained on Llama-2-7b and GPT2-XL representations exhibit comparable performance levels. Both models show a slight decrease in generalization and locality performance as the number of edits increases from 1000 to 2000, with minimal decline after that.

Examination of projection network behaviour reveals interesting patterns in generalization and locality failures based on the varying distances between training edits and their respective paraphrases and irrelevant prompts after the training of the projection network. The varying distances result in different thresholds for each edit, which can cause errors when the closest edit to an irrelevant prompt example has a high threshold. To quantify these observations, we employed ROUGE scores in a comparative study of generalization outcomes. Appendix G provides further analysis of the learned projection space.

| Model | NLI | Sentiment | Summarization |
|---|---|---|---|
| Llama-2-7b | 0.6476 | 0.6573 | 0.1865 |
| Llama-2-7b$_{PENME}$ | 0.6428 | 0.6573 | 0.1865 |
| GPT2-XL | 0.5128 | 0.4630 | 0.0936 |
| GPT2-XL$_{PENME}$ | 0.5128 | 0.4630 | 0.0936 |

*Table 2.* Downstream task performance of GPT2-XL and Llama-2-7B before and after PENME editing, demonstrating the method's effect on model general capabilities.

| Model | Fluency | Reference Score |
|---|---|---|
| Llama-2-7b | 611.54 | 16.57 |
| Llama-2-7b$_{PENME}$ | 622.36 | 21.98 |

*Table 3.* Evaluation of long-form generation by Llama-2-7B pre- and post-editing with PENME (via IKE).

## 7. Generalization and Locality

To demonstrate the trade-off between generalization and locality, we conducted an ablation study by varying the $\tau$ parameter, which modulates the similarity threshold defining an edit's scope. Figure 5 presents the results for GPT2-XL and T5-small. The trends observed for GPT2-XL and Llama-2-7b are similar. Therefore, for clearer visualization, we present the detailed results for Llama-2-7b separately in Appendix F.1. Setting a low $\tau$ value achieves near-perfect locality but poor generalization. As we incrementally increase the threshold, generalization improves while locality declines gradually. Each model exhibits an optimal threshold where generalization and locality are balanced; these thresholds can be adjusted to suit specific use cases e.g. high locality to ensure no degradation in the original model.

Figure 5 also illustrates the impact of varying the similarity threshold for edit-to-edit pairings in the training dataset on the projector network's learning. Edit-to-edit pairings $\phi$ which move edits farther away from each other are central to training a robust projector network. The threshold value for edit-to-edit pairings $\phi$ significantly impacts training stability and performance. Higher thresholds, such as 0.75, result in fewer pairings and lead to unstable training for both Llama-2-7b and GPT2-XL models, ultimately resulting in poor performance. Conversely, lower thresholds, exemplified by 0.6, increase the number of pairings and enhances stability.

## 8. Downstream Task

Table 2 reports the performance of GPT2-XL and Llama-2-7B across downstream NLP tasks, before and after applying PENME. For GPT2-XL, performance remains unchanged post-edit, indicating that the edit was successfully localised without negatively affecting general capabilities. Similarly, Llama-2-7B exhibits stable performance on summarization and classification tasks, with only a minor drop observed on the NLI task. These results suggest that PENME effectively preserves model general capabilities.

## 9. Long Form Generation

As discussed in Section 4, vector playback or trainable LoRA blocks can be used to support long-form generation. In this section, we adopt the retrieval-based prompting strategy introduced by IKE (Zheng et al., 2023) to perform long form generation. Since PENME operates on early layers of the model, inference can be halted early when user input falls within the scope of an edit. In such cases, the edited information is retrieved and used to construct a new prompt, combining the retrieved content with the user query to guide the model's generation.

To evaluate this approach, metrics which include Generation Entropy (Fluency) and Consistency (Reference score) (Meng et al., 2023) are used. Table 3 presents the results of this approach on the CounterFact dataset. We observe an improvement in fluency of the model's generation (fluency), indicating a reduction in repetitive or redundant output, and a corresponding increase in reference score, reflecting better factual alignment. Sampled generations and detailed analysis along with a discussion on multi-hop editing, is provided in Appendix I. The samples demonstrate that the edited fact is consistently preserved and integrated across the full output.

## 10. Conclusion

In this paper, we raised awareness of a critical vulnerability in weight-preserving adapter-based model editing techniques: lexical bias in the representation space. We developed a projection-based method PENME trained via contrastive learning to disentangle lexical and semantic similarity which originally would cause misfiring on irrelevant prompts with a high lexical overlap. Empirical evaluations showed PENME's superior performance across varying levels of task complexity. On the zsRE dataset, it achieved impressive generalization and locality scores exceeding 0.90, demonstrating that our method is satisfactorily able to balance generalization and locality using distance metrics in this new projected space. Notably, when assessed on the more challenging Counterfact benchmark, the system maintained robust performance, attaining scores above 0.80 for both generalization and locality metrics. This performance on Counterfact is particularly significant given the benchmark's increased difficulty, underscoring PENME's efficacy. In future work, we aim to investigate whether a projector pre-trained on a large-scale dataset can serve as a plug-and-play component for cross-lingual generalization. Additionally, we plan to explore whether the projector can be trained and updated incrementally with new edits, thereby reducing training overhead and improving scalability.

## Impact Statement

This paper advances model editing by mitigating lexical bias in adapter-based approaches, enabling precise and targeted updates to language models. While our method does not introduce an additional ethical risk beyond those already associated with language models, the model editing techniques, in general, can be exploited to inject unsafe behaviour into a model (Li et al., 2024c).

## Acknowledgment

We acknowledge the support of the Natural Sciences and Engineering Research Council of Canada (NSERC), Canada Foundation of Innovation (CFI), and Research Nova Scotia. Advanced computing resources are provided by ACENET, the regional partner in Atlantic Canada, and the Digital Research Alliance of Canada.

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

# A. Data Construction and Inference for PENME

The projection network is similar to the feed-forward layers in a transformer as it contains two layers with ReLU activation in between, with the addition of the Batch Normalization layer, a common element in contrastive learning. The network is trained via contrastive learning, which requires a dataset based on a pair of inputs with positive and negative labels. The algorithm 1 data construction process.

At runtime, upon receiving a user query, PENME checks if the query falls within the editing scope of the edits. If so, the new output from memory is retrieved and the inference process is stopped. Alternative generation mechanism is discussed in Section 4 of the main paper text, which includes replacing the storage of the new fact as text with playback vectors or LoRa block indices. Additionally, using the retrieved information in context learning based generation can be used. The inference pipeline for PENME is given in 2.

---

**Algorithm 1** Data Construction for Projector Network

---

1: **Input:** $num\_overall\_negative$, $threshold\_edit\_pairings$
2: **Input:** memory $\leftarrow \{\}$           {Memory storage}
3: **Input:** dataset_pairs $\leftarrow []$
4: **Input:** $Cos(\cdot, \cdot)$           {Cosine similarity function}
5: **Input:** dataset rows $r_i = (x_i, y_i, \{p_{ij}\}, \{p_{ij}^{\neg}\})$
6: **for** each $r_i$ in dataset **do**
7:     **for** each $p_{ij}$ and $p_{ij}^{\neg}$ in $r_i$ **do**
8:         Add $(x_i, p_{ij})$ to dataset_pairs         {Positive pair}
9:         Add $(x_i, p_{ij}^{\neg})$ to dataset_pairs         {Negative pair}
10:     **end for**
11:     **for** each $r_t$ in dataset, where $i \neq t$ **do**
12:         **if** $Cos(x_i, x_t) > threshold\_edit\_pairings$ **then**
13:             Add $(x_i, x_t)$ to dataset_pairs         {Negative edit-to-edit pair}
14:         **end if**
15:         **for** each $p_{tj}^{\neg}$ in $r_t$ **do**
16:             Add $(Cos(x_i, p_{tj}^{\neg}), (x_i, p_{tj}^{\neg}))$ to memory
17:         **end for**
18:     **end for**
19: **end for**
20: Sort memory in descending order by similarity
21: Add top-$num\_overall\_negative$ items from memory to dataset_pairs
22: **return** dataset_pairs

---

**Algorithm 2** Inference for LLM with PENME

---

1: **Input:** $h_l(\cdot)$           {LLM output at layer $l$}
2: **Input:** $g(\cdot)$           {Projector network}
3: **Input:** $D(\cdot, \cdot)$           {Euclidean distance}
4: **Input:** codebook = $\{k_i: (v_i^{\delta}, v_i^{\mu})\}$
5: **Input:** user prompt $x_t$
6: $h_l \leftarrow h_l(x_t)$
7: $a_x \leftarrow g(h_l)$
8: Find $k^* = \arg\min_{k_i} D(a_x, k_i)$
9: **if** $D(a_x, k^*) < v_{k^*}^{\delta}$ **then**
10:     **return** $y_{k^*}$
11: **else**
12:     **return** $y$           {Fallback to base LLM output}
13: **end if**

---

# B. Paraphrases and Irrelevant prompts Distance Analysis

Table 4 shows the distance between edits and their respective paraphrases and irrelevant prompts across various measurement metrics. From the distances the average paraphrase distance (AvgPD) and average distances between training and test paraphrases (AvgDTTP), we can see that they are generally a little farther than the test paraphrases and are on average a bit farther from the edit than train paraphrases. On the other hand, the average irrelevant prompt distance (AvgPN) and average distances between training and test irrelevant prompts (AvgDTTN) show that the test irrelevant prompts are a little closer to the edit as compared to the train irrelevant prompts.

# C. Comparison Scoping Mechanism: PENME versus MELO and GRACE

To demonstrate the improvement in inference time for selecting the appropriate key, we compare PENME with MELO across various sample sizes of edits, ranging from 50 to 300 in increments of 50 shown in table 5. The results show that PENME outperforms MELO in terms of speed and also highlight the number of keys forgotten during training due to the design of its scoping mechanism, as well as the number of entries for which the radius had to be reduced.

# D. Experimentation and Implementation Details

### D.1. Experimentation Setup

For our comparative analysis, we contrast against baseline methods such as simple fine-tuning (FT), alongside advanced approaches drawn from relevant literature. These encompass GRACE (Hartvigsen et al., 2023; Yu et al., 2024), employing adapter-based editing with a similarity-based scoping mechanism. SERAC (Mitchell et al., 2022), a multimodal editing approach incorporating a scoping classifier, memory database, and counterfactual model alongside the target model and MEMIT (Meng et al., 2023) an editing approach designed for decoder only model adopts a model-editing strategy by identifying and updating knowledge-contained model layers' weight matrices.

In evaluating our approach, we adhere to the metrics outlined in §3. Regarding generalization, we define a paraphrase as generalized if it aligns with the correct edit and falls below its distance threshold. For assessing locality, we maintain that locality is preserved when the distance between matched edits exceeds its threshold. Any other instances are categorized as misfires. It is important to note that (Hartvigsen et al., 2023; Yu et al., 2024) utilize token F1 Accuracy and (Mitchell et al., 2022) use a metric based on token probabilities. These metrics are softer in nature which allows for higher scores.

### D.1.1. COMPUTATION RESOURCES

Training for all projector networks is conducted on an NVIDIA P100 GPU with 16GB VRAM. A larger VRAM or RAM capacity is only necessary for the initial extraction of layer representations from the pre-trained language models. For the evaluation of approaches from relevant literature, some of which demanded greater computational resources, we employed NVIDIA A100 GPUs with 40GB and 80GB VRAM. All editing approaches were supported are implemented using the default configurations provided in the Easy-Editor library (Wang et al., 2023). It is important to note that not all models are supported across all editing methods. For instance, Llama-2-7b is not supported for MELO. For some models such as T5-small, limited support is provided therefore, we utilise the code provided by the paper's authors.

### D.1.2. HYPERPARAMETERS

For training the projector networks, we utilise the Adam optimiser. we experiment with various learning rates $1e^{1-2}, 2e^{1-2}, 3e^{1-2}$. we find that a moderate learning rate is required to learn faster while not overfitting, hence we choose $1e^{1-2}$, with a learning rate decay rate of $0.01$. All projection networks are trained for 200 epochs using a batch size of 8192 and an early stopping patience of 8 epochs. For selecting the margin $m$ in the contrastive learning cost function we ablate on the hyperparameter m for the GPT2-XL model. The table 6 shows the margin m along with the adjustment to $\tau$ for balanced results for generalization and locality. It can be observed from the table to achieve high-performance minimum value of 30 needs to be utilized. The higher the the value for $m$ the better the score for localization. The value chosen is 40 which has the most balanced results.

*Table 4.* Distance analysis of distances between edit and its respective paraphrase and irrelevant prompts. The metrics for measurement include average/max/min paraphrase distance (AvgPD)(MaxPD)(MinPD), average/max/min irrelevant prompts distance (AvgND),(MaxND)(MinND), average/max/min distances between training and test paraphrase (AvgDTTP)(MaxDTTP)(MinDTTP), the average distance between farthest edit and closest irrelevant prompt (AvgCPFN), and average/max/min distances between training and test irrelevant prompts (AvgDTTN)(MaxDTTN)(MinDTTN).

| Model | Measurement Metric | Training Set | Test Set | Train vs Test |
|---|---|---|---|---|
| | AvgPD | 0.240 | 0.254 | - |
| | MinPD | 0.0 | 0.02 | - |
| | MaxPD | 0.829 | 1.59 | - |
| | AvgND | 1.436 | 1.379 | - |
| | MinND | 0.803 | 0.616 | - |
| | MaxND | 1.884 | 1.853 | - |
| Llama-2-7b | AvgCPFN | 0.348 | 0.893 | - |
| | AvgDTTP | - | - | 0.013 |
| | MaxDTTP | - | - | 1.459 |
| | MinDTTP | - | - | -0.634 |
| | AvgDTTN | - | - | -0.227 |
| | MaxDTTN | - | - | -1.130 |
| | MinDTTN | - | - | 0.0 |
| | AvgPD | 0.409 | 0.491 | - |
| | MinPD | 0.0 | 0.002 | - |
| | MaxPD | 1.375 | 1.381 | - |
| | AvgND | 0.468 | 0.534 | - |
| | MinND | 0.005 | 0.010 | - |
| | MaxND | 1.384 | 1.386 | - |
| T5-small | AvgCPFN | 0.193 | 0.238 | - |
| | AvgDTTP | - | - | 0.018 |
| | MaxDTTP | - | - | 1.273 |
| | MinDTTP | - | - | -1.290 |
| | AvgDTTN | - | - | -0.276 |
| | MaxDTTN | - | - | -1.341 |
| | MinDTTN | - | - | 0.0 |
| | AvgPD | 0.378 | 0.349 | - |
| | MinPD | 0.0 | 0.01 | - |
| | MaxPD | 1.49 | 1.395 | - |
| | AvgND | 1.174 | 1.092 | - |
| | MinND | 0.227 | 0.368 | - |
| | MaxND | 1.709 | 1.728 | - |
| GPT2-XL | AvgCPFN | 0.382 | 0.700 | - |
| | AvgDTTP | - | - | 0.008 |
| | MaxDTTP | - | - | 1.368 |
| | MinDTTP | - | - | -1.046 |
| | AvgDTTN | - | - | -0.148 |
| | MaxDTTN | - | - | -0.856 |
| | MinDTTN | - | - | 0.0 |

*Table 5.* Runtime Performance Comparison of PENME versus MELO. For PENME, the number of Codebook entries is the same as the number of edits.

| Number of Edits | PENME | MELO/GRACE | | | |
|---|---|---|---|---|---|
| | Runtime (ms) | Runtime (ms) | Codebook Entries | Edits Forgotten | Edit Conflict |
| 50 | 0.024 ± 0.003 | 0.316 ± 0.090 | 269 | 24 | 21 |
| 100 | 0.115 ± 0.129 | 0.364 ± 0.050 | 523 | 77 | 66 |
| 150 | 0.188 ± 0.182 | 0.624 ± 0.082 | 785 | 132 | 114 |
| 200 | 0.279 ± 0.170 | 1.423 ± 0.180 | 1048 | 188 | 169 |
| 250 | 0.404 ± 0.170 | 1.681 ± 0.205 | 1319 | 254 | 217 |
| 300 | 0.418 ± 0.125 | 2.149 ± 1.069 | 1554 | 301 | 268 |

*Table 6.* The table shows how the performance changes along with the required threshold adjustment to $\tau$ as margin $m$ in contrastive loss is changed

| Margin $m$ | Threshold Adjustment $\tau$ | Generalization | Locality |
|---|---|---|---|
| 10 | 0 | 0.634 | 0.831 |
| 20 | 3 | 0.891 | 0.880 |
| 30 | 6 | 0.958 | 0.948 |
| 40 | 8 | 0.967 | 0.977 |
| 50 | 11 | **0.978** | 0.965 |
| 60 | 13 | 0.976 | 0.986 |
| 70 | 17 | 0.973 | 0.976 |
| 80 | 17 | 0.973 | 0.976 |
| 90 | 20 | 0.928 | **0.986** |

## D.2. Data Processing

**Counterfact:** Each row in the Counterfact consists of an edit prompt, two paraphrase prompts, multiple irrelevant prompts and an edit label $x_i, y_i, [p_1, p_2], [p_{i1}^{\rightharpoonup}...p_{ij}^{\rightharpoonup}])$. For the training dataset, we extract the edit prompt $x_i$, one randomly sampled paraphrase $p_i$ and half the irrelevant prompts $p_{ij}^{\rightharpoonup}$. For creating additional paraphrases for the training set we utilize the extracted edit prompt and paraphrase prompt as input to ChatGPT and use it to generate three additional paraphrases for training. We ensure that the generated paraphrase follows the $(s, r, o^*)$ triplet format that the dataset uses. The test set for locality and generalization compromises of the paraphrase and irrelevant prompts not sampled from the training set.

**zsRE:** The zsRE dataset comprises of rows containing a sample question, its corresponding new label, and multiple rephrased questions along with its filtered rephrased questions. We constructed this dataset following methodologies established in the relevant literature. A balanced subset of paraphrases are derived from the filtered rephrased questions for training and testing purposes. For irrelevant prompts samples, we randomly selected an equal number of questions from the NQ dataset for training and testing while ensuring no overlap in questions.

To highlight the lexicality issue in the datasets, we compute several token overlap metrics between pairs of (edits, paraphrases) $(x_i, p_{ij})$ and (edits, irrelevant prompts) $(x_i, p_{ij}^{\rightharpoonup})$. The results are presented in Table 7 and dataset samples in Table 8. From the token overlap metrics table, it is evident that the edit prompt and irrelevant prompts show high overlap in Counterfact, whereas the overlap is minimal in zsRE. This, coupled with the experiment in Section 6.1, highlights the significant challenges observed in the Counterfact dataset.

*Table 7.* Comparison between zsRE and Counterfact for token overlap metrics

| | | zsRE | | | | Counterfact | | | |
|---|---|---|---|---|---|---|---|---|---|
| Metric | Pair Type | Score | Precision | Recall | F1 | Value | Precision | Recall | F1 |
| Jaccard Similarity | $(x_i, p_{ij})$ | 0.399 | - | - | - | 0.401 | - | - | - |
| Jaccard Similarity | $(x_i, p_{ij}^{\neg})$ | 0.086 | - | - | - | 0.430 | - | - | - |
| ROUGE-1 | $(x_i, p_{ij})$ | - | 0.321 | 0.315 | 0.316 | - | 0.310 | 0.325 | 0.307 |
| ROUGE-1 | $(x_i, p_{ij}^{\neg})$ | - | 0.076 | 0.087 | 0.079 | - | 0.295 | 0.293 | 0.290 |
| ROUGE-2 | $(x_i, p_{ij})$ | - | 0.189 | 0.194 | 0.194 | - | 0.189 | 0.198 | 0.184 |
| ROUGE-2 | $(x_i, p_{ij}^{\neg})$ | - | 0.008 | 0.008 | 0.008 | - | 0.205 | 0.203 | 0.201 |
| ROUGE-L | $(x_i, p_{ij})$ | - | 0.299 | 0.294 | 0.293 | - | 0.299 | 0.312 | 0.295 |
| ROUGE-L | $(x_i, p_{ij}^{\neg})$ | - | 0.070 | 0.080 | 0.073 | - | 0.294 | 0.292 | 0.289 |

| | Counterfact | | | zsRE | | |
|---|---|---|---|---|---|---|
| Edit | Paraphrase | Neighbour | Edit | Paraphrase | Neighbour NQ dataset |
| The twin city of Cologne is | What is the twin city of Cologne? It is | The twin city of London is | Which river system contains Laborec? | What river system does Laborec contain? | Where does the last name serrano come from? |
| Alexander Zinoviev works in the area of | Alexander Zinoviev's domain of work is | TFred W. Riggs works in the area of | Which airport does Air Seychelles operate in? | Which airport is closely linked to Air Seychelles? | How many students attend chippewa valley high school? |
| The original language of Kondura was | The language of Kondura is | The language of Taal is | The country of origin for Kala Pul is what? | Which was the country for Kala Pul? | "When do the new sky sports channels launch? |
| Thomas Arne died in the city of | Thomas Arne lost their life at | Bill Brandt died in the city of | What label was responsible for Wild World? | What was the label Wild World? | Who composed the music for avengers infinity war? |

*Table 8.* Random samples from the Counterfact and zsRE datasets.

# E. Projector Network and Lexical Bias

## E.1. Lexical Dominance Layer Analysis

Figure 7 shows the percentage of edits samples where irrelevant prompts were closer to the edits for all models across all layers.

## E.2. Layer-Wise Analysis of the Projector Network

Figure 8 shows the results for generalization and locality for the T5-small model. The results suggest that performance remains largely consistent; however, training tends to require more time to converge at higher layers.

# F. Visualizations

## F.1. GENERALIZATION AND LOCALITY for Llama-2-7b

Figure 9 shows generalization and locality trade-off as a function of varying distance thresholds $\tau$ and $\phi$ for the Llama-2-7b model.

## F.2. PCA

Figures 10 and 11 present the two-dimensional PCA of the model representations and projector network representations for the Llama-2-7b and GPT2-XL models, respectively. The visual demonstrates that irrelevant prompts are closely aligned with edit prompts, while edit prompts also show proximity to other edit prompts within the original model representations. The projector network, however, effectively mitigates this effect by learning a disentangled representation space.

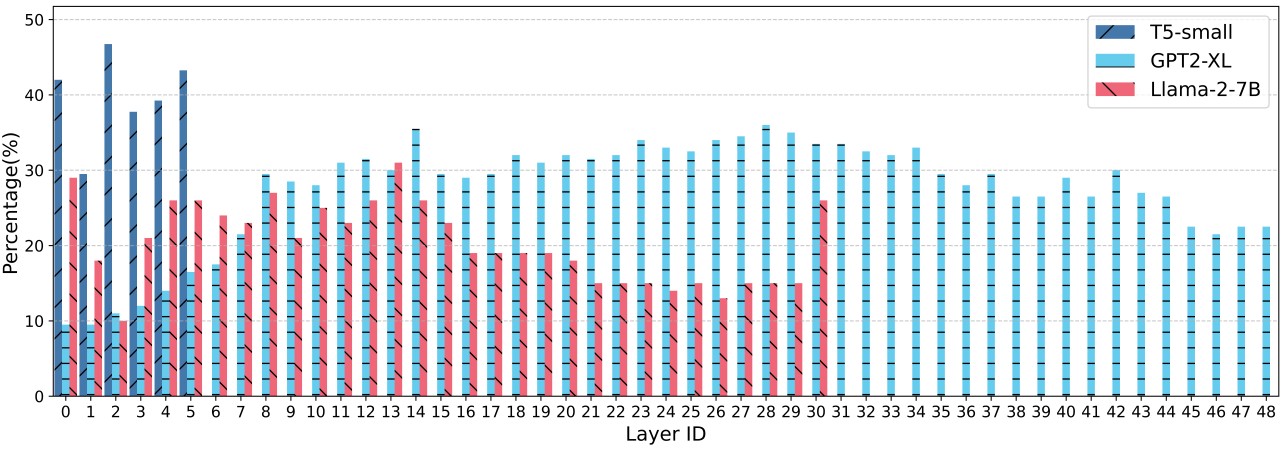

*Figure 7.* Percentage of samples *where edits are closer* to irrelevant prompts as compared to paraphrases in the representations space of different models across all layers. T5-small, GPT2-XL and Llama-2-7b have 6, 32, 48 layers, respectively.

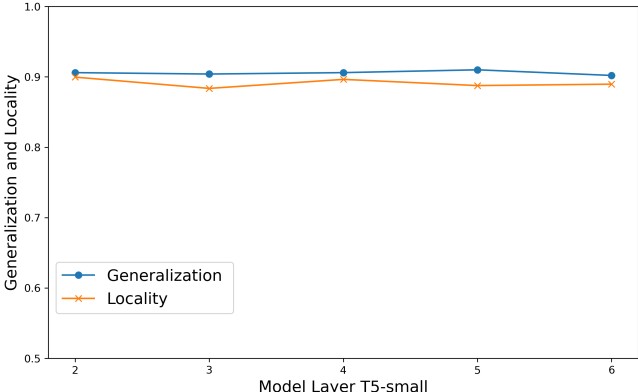

*Figure 8.* Generalization and locality scores for various projector networks trained on layers of T5-small using 500 samples from Counterfact.

## G. Error Analysis Projector Network

To investigate the reasons behind failures in PENME, we performed a comprehensive error analysis across our models. Our findings indicate that contrastive learning significantly mitigates lexical bias. However, due to the inherent variability in lexical pattern distribution within the dataset, there remains potential for further optimization in the projection phase.

The training process of the projector network does not lead to uniform distances between each edit, its paraphrases and irrelevant prompts for all samples. This paired with individually varying thresholds for edits leads to misfires. To illustrate this problem, we format the results of each dataset sample for automatic inspection. For all paraphrases and irrelevant prompts in the test set, we extract the nearest key/edit, the ground truth edit/key, the distance to the nearest key/edit, and the distance to the ground truth edit/key. Table 9 shows rouge scores (Lin, 2004) for two possible scenarios i.e. success and failure of generalization and locality. We also show separately the score for where generalization failure occurs due to distance not meeting the set threshold. Moreover, since failures can occur in similarities with unrelated edits we show locality and paraphrase failure with both ground truth edit and matched edit.

For cases of successful generalization, we observe a substantial uni-gram overlap and a moderate bi-gram overlap between the edited sentences and their paraphrases. The ROUGE-L scores are similarly high for these metrics, indicating that the sentences likely share similar tokens in the same sequence. This implies that the attention mechanism produces similar representations, leading to a high degree of similarity. For locality success, we can see that although there is significant token overlap between irrelevant prompts and their target edits, the irrelevant prompts had higher similarity with some

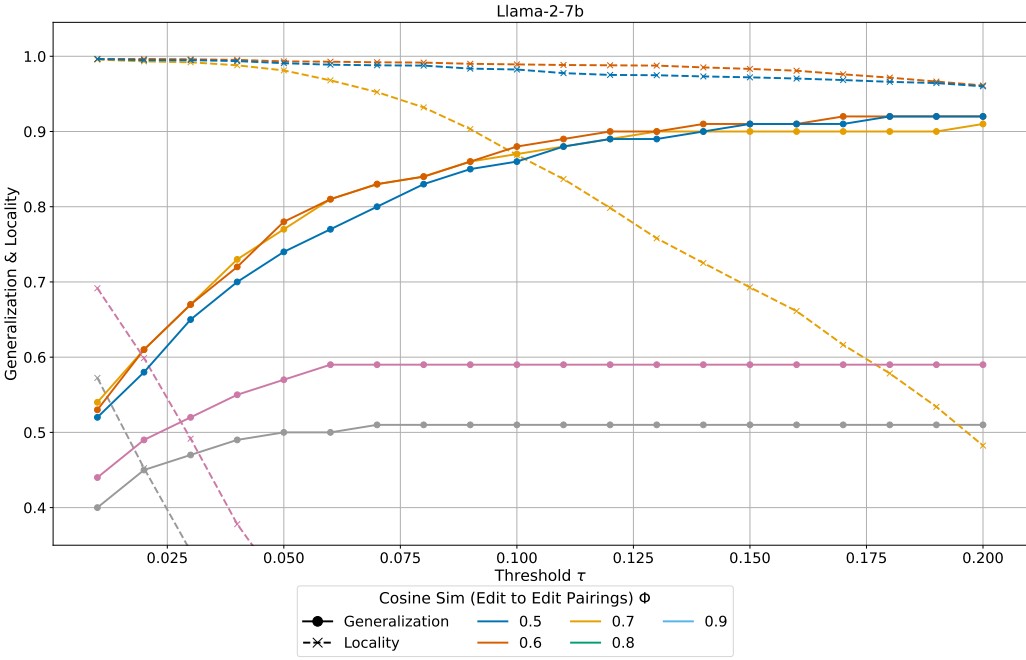

*Figure 9.* Generalization and Locality trade-off a function of varying distance thresholds $\tau$ and $\phi$

other edits with low token overlap, this means our approach of pushing irrelevant prompt sentences farther away is able to generalize to unseen irrelevant prompts.

In cases of generalization failure, the ROUGE scores for paraphrases compared with the ground truth are slightly lower than those observed in successful instances. Although there is some token overlap with the target edits, the matched edits exhibit even less token overlap. On the other hand for locality failure, we can see that the prediction case token overlap is higher as compared to locality success, moreover, the overlap is higher as compared to ground truth edits. Thus lexicality based similarity is not the issue but rather the varying thresholds, which in some cases are large leads to misfires.

## H. Details Downstream Tasks

For the evaluation of sentiment classification and natural language inference (NLI), we randomly sample 2,000 examples from their respective datasets. For the summarization task, we use 1,000 randomly selected examples. The prompts used for evaluation are listed in Table 10. For both the sentiment classification and summarization tasks, providing a single in-context example was necessary to achieve reasonable model performance.

## I. Details Long Form Generation

Sampled generations using Llama-2-7b with max token length set to 300 are presented in Table 11.

We use the following prompt, which is stored as the value in the codebook:

```
You have a new fact:  {edit prompt}.
Based on this fact, complete the following sentence to answer the question:
{query}
Your answer should specifically incorporate the new fact I've shared.
Paragraph:  {query}
```

The generated text shows that the edited fact is propagated throughout the generation, and the generation is coherent. However, it is important to note that IKE-based generation, as well as by other approaches, including weight-modifying methods, do not guarantee the elimination of hallucinated content that may accompany the edited information. Although

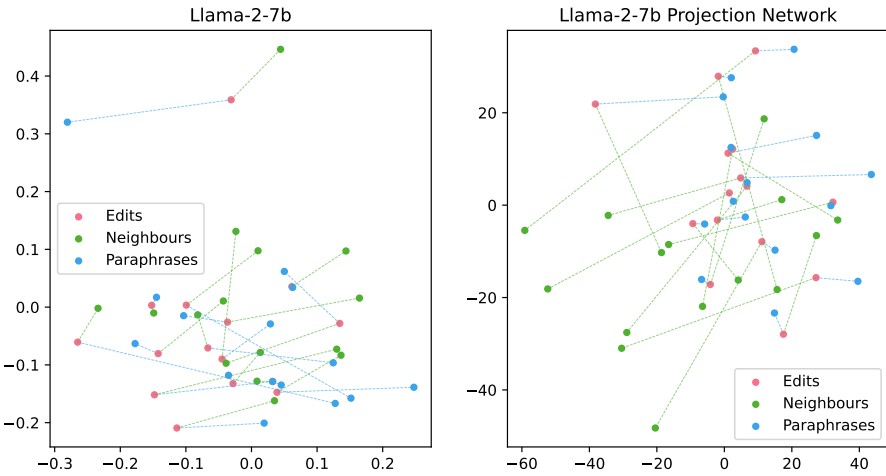

*Figure 10.* Generalization and locality scores for various projector networks trained on layers of T5-small using 500 samples from Counterfact. The lines show edits and a respective paraphrase and irrelevant prompts.

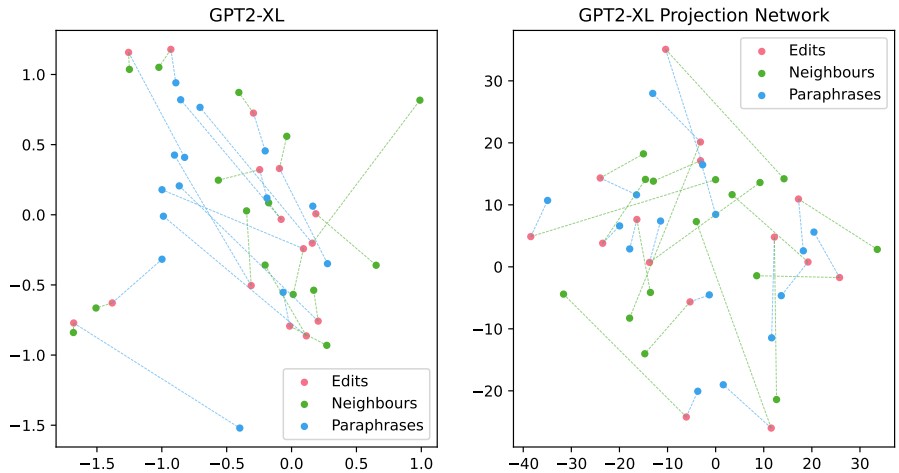

*Figure 11.* Two dimensional PCA on GPT2-XL model representation and the trained projector network.

the target fact is often correctly inserted, unrelated or inaccurate details can still be produced. Addressing this limitation remains an open challenge and a promising direction for future work.

For multi-hop editing, prompts whose outputs are expected to change can be passed through the model and projector to generate corresponding keys. These keys can then be linked to a central edit, enabling the model at runtime to retrieve and associate relevant information with the original edit. Through in-context reasoning, the model can subsequently produce the desired output

## J. Limitations

Training the projection network in PENME using the contrastive learning scheme is sensitive, requiring tuning of hyper-parameters such as the learning rate and contrastive loss margin. Effective network training also hinges on the careful construction of training data, which requires consideration of edit-to-edit pairings. Finally, the thresholds for the codebook-based retrieval system, though dynamically determined from training data, can vary across different models, necessitating adjustments to the alpha ($a$) parameter for each model.

*Table 9.* ROUGE Evaluation Scores

| Model | Rouge-1 2.5% CI | Rouge-1 97% CI | Rouge-2 2.5% CI | Rouge-2 97% CI | RougeL 2.5% CI | RougeL 97% CI |
|---|---|---|---|---|---|---|
| **Generalization Success** | | | | | | |
| T5-small | 1.00 | 0.95 | 1.05 | 0.706 | 0.65 | 0.75 |
| Llama-2-7b | 0.629 | 0.639 | 0.382 | 0.394 | 0.608 | 0.619 |
| GPT2-XL | 0.655 | 0.666 | 0.403 | 0.417 | 0.642 | 0.653 |
| **Generalization Failure (prediction)** | | | | | | |
| T5-small | 1.00 | 0.95 | 1.05 | 0.706 | 0.65 | 0.75 |
| Llama-2-7b | 0.133 | 0.173 | 0.056 | 0.091 | 0.125 | 0.162 |
| GPT2-XL | 0.122616 | 0.160 | 0.056 | 0.090 | 0.117 | 0.153 |
| **Generalization Failure (ground truth)** | | | | | | |
| T5-small | 1.00 | 0.95 | 1.05 | 0.706 | 0.65 | 0.75 |
| Llama-2-7b | 0.488 | 0.518 | 0.270 | 0.296 | 0.460 | 0.489 |
| GPT2-XL | 0.501 | 0.527 | 0.284 | 0.310 | 0.474 | 0.500 |
| **Locality Success (prediction)** | | | | | | |
| T5-small | 0.100 | 0.104 | 0.011 | 0.013 | 0.096 | 0.099 |
| Llama-2-7b | 0.100 | 0.104 | 0.011 | 0.013 | 0.096 | 0.099 |
| GPT2-XL | 0.095 | 0.100 | 0.011 | 0.013 | 0.092 | 0.095 |
| **Locality Success (ground truth)** | | | | | | |
| T5-small | 0.100 | 0.104 | 0.011 | 0.013 | 0.096 | 0.099 |
| Llama-2-7b | 0.487 | 0.518 | 0.269 | 0.296 | 0.459 | 0.489 |
| GPT2-XL | 0.176 | 0.217 | 0.036 | 0.059 | 0.173 | 0.211 |
| **Locality Failure (prediction)** | | | | | | |
| T5-small | 0.566 | 0.577 | 0.390 | 0.403 | 0.562 | 0.574 |
| Llama-2-7b | 0.259 | 0.277 | 0.148 | 0.164 | 0.247 | 0.264 |
| GPT2-XL | 0.254 | 0.273 | 0.147 | 0.164 | 0.244 | 0.262 |
| **Locality Failure (ground truth)** | | | | | | |
| T5-small | 0.203 | 0.212 | 0.052 | 0.058 | 0.197 | 0.206 |
| Llama-2-7b | 0.201 | 0.206 | 0.049 | 0.053 | 0.195 | 0.201 |
| GPT2-XL | 0.207 | 0.218 | 0.052 | 0.059 | 0.201 | 0.212 |
| **Generalization Distance Failure** | | | | | | |
| T5-small | 1.00 | 0.95 | 1.05 | 0.706 | 0.65 | 0.75 |
| GPT2-XL | 0.522 | 0.551 | 0.279 | 0.309 | 0.484 | 0.512 |
| Llama-2-7b | 0.495 | 0.579 | 0.252 | 0.324 | 0.455 | 0.529 |

| Model | NLI | Sentiment Classification | Summarization |
|---|---|---|---|
| Llama-2-7b | {sentence1} entails the {sentence2}. True or False? Answer: | Choose from one of these: anger, fear, joy, love, sadness, surprise. Example: {example} Text: {text} Sentiment: | Example: {example} Now generate short summary of the following: Article: {article} Summary: |
| GPT2-XL | {sentence1} entails the {sentence2}. True or False? Answer: | Choose from one of these: anger, fear, joy, love, sadness, surprise. Example: {example} Text: {text} Sentiment: | What is the main takeaway from the following article? {text} Summary: |

*Table 10.* Prompt used for GPT2-XL and Llama-2-7b for downstream tasks.

*Table 11.* Example of an edited factual prompt and generated output.

| Prompt Edit | Original Information | Edit | Query | Output |
|---|---|---|---|---|
| What is the twin city of Shanghai? It is | Barcelona | Dresden | The twin city of Shanghai is none other than | The twin city of Shanghai is none other than **Dresden** Germany. This may come as a surprise to many as **Dresden** is located in Eastern Germany while Shanghai is a major city in Eastern China. However, the two cities have a long history of cultural and economic ties dating back to the 19th century when both were part of the German Empire. In the years following World War II, **Dresden** and Shanghai maintained close diplomatic relations, and in 1981 the two cities officially designated each other as twin cities. Today, the connection between **Dresden** and Shanghai continues to flourish with regular exchanges in the fields of education, culture, and trade. |
| Baal Shem of London speaks the language | Hebrew | French | Baal Shem of London is proficient in | Baal Shem of London is proficient in multiple languages including **French**. This is a remarkable feat given that he is a spiritual leader and healer who is said to have the ability to communicate with the divine. However his proficiency in **French** is particularly noteworthy as it is a language that is not commonly associated with his spiritual tradition. Despite this Baal Shem of London has demonstrated a remarkable ability to master this language allowing him to connect with and heal individuals from a diverse range of cultural backgrounds. This is yet another testament to his incredible spiritual abilities and his commitment to serving others. |

