# OpenReview forum: "Resolving Lexical Bias in Model Editing"
_ICML.cc/2025/Conference — ICML 2025 poster_

### Official Review · Reviewer_s3vv · 2025-03-10

**Overall Recommendation:** 4

**Summary:**

This paper addresses the challenge of editing the outputs of large language models without degrading their overall performance. Traditional methods directly modify model weights, often causing undesirable side effects. In contrast, recent approaches use adapters that trigger edits based on semantic similarity in the representation space. However, these adapter methods are shown to be susceptible to strong lexical biases—resulting in unintended edits for prompts with overlapping words. To overcome this, the paper introduces a principled method for learning a disentangled representation space. This new space allows for the precise localization of edits by keeping unrelated prompts distant while keeping semantically equivalent (or paraphrased) prompts close together. The proposed method, Projector Editor Networks for Model Editing (PENME), achieves state-of-the-art editing performance. It is not only more computationally efficient during inference than previous approaches but also adaptable across various architectures.

**Claims And Evidence:**

The claims made  in the submission are supported by the experiments.

**Essential References Not Discussed:**

No.

**Experimental Designs Or Analyses:**

The experimental designs are sound.

**Methods And Evaluation Criteria:**

The proposed methods make sense for the problem.

**Other Comments Or Suggestions:**

No.

**Other Strengths And Weaknesses:**

Strengths:

By learning a disentangled representation space, the method effectively differentiates between relevant and irrelevant prompts, reducing unintended modifications.

PENME offers faster inference times compared to prior methods, making it more practical for real-world applications.


The paper identifies and mitigates the vulnerability of previous adapter methods to strong lexical biases, enhancing reliability. The approach is designed to work across different model architectures, increasing its applicability.

Weaknesses:

Although adaptable across architectures, the paper may not fully explore how the approach scales with increasingly larger and more complex models.

The success of precise edit localization hinges on the quality of the learned disentangled representation, which might require extensive tuning and may not generalize perfectly across all domains.

**Questions For Authors:**

See weaknesses.

**Relation To Broader Scientific Literature:**

This paper demonstrates that current adapter methods are critically vulnerable to strong lexical biases, which can result in edits being applied to irrelevant prompts that share overlapping words. This paper introduces a principled approach to learning a disentangled representation space, enabling the precise localization of edits by keeping unrelated prompts distinct while preserving the proximity of semantically similar paraphrases.

**Theoretical Claims:**

This paper does not contain theoretical claims.

---

> ### Author Rebuttal · Authors · 2025-04-01
>
> **Comment: Although adaptable across architectures, the paper may not fully explore how the approach scales with increasingly larger and more complex models**
>
> The proposed approach operates at a single layer of the model, making it lightweight and efficient, with minimal dependence on model size. Scalability is not a major concern, as the method avoids full model retraining and extensive parameter updates. In our experiments, we employed a direct output strategy by modifying the architecture to support multi-output generation. However, generation control mechanisms such as playback vectors and lightweight LoRA blocks have also demonstrated robust performance and can serve as effective alternatives to direct output. These design choices suggest that the approach can scale well to larger models or more complex generation tasks without incurring significant computational overhead.
>
> **Comment: The success of precise edit localization hinges on the quality of the learned disentangled representation, which might require extensive tuning and may not generalize perfectly across all domains.**
>
> Our analysis identifies the key factor contributing to locality challenges in the model’s representation space, which is lexical bias.  We have evaluated PENME’s generalization capabilities by assessing performance transfer from the CounterFact to the zsRE dataset. While our current experiments focus on general QA data, the results suggest that PENME can adapt to new domains with minimal supervision. We hypothesize that for specialized domains, PENME can generalize with a small amount of in-domain data.

---

### Official Review · Reviewer_cuog · 2025-03-10

**Overall Recommendation:** 2

**Summary:**

This paper propose a method for model editing, following GRACE paper. The authors witnessed that the model intermediate representations are ambiguous to distinguish irrelevant prompts and paraphrases given an editing prompt. The authors adopt the contrastive learning technique to well separate them, thus enforcing better locality and generalizability.

**Claims And Evidence:**

There is no definition given for Lexical Bias in the paper.

The evidence presentation is a bit messed up. For example,

1. Figure 2 uses "Irrelevant Prompts" while Figure 3 uses "Neighbors". There is no definition for "neighbors" throughout the paper
2. The bars overlap with each other in Figure 4
3. The legends for color do not match those lines in Figure 5. There is no green line or clear blue line
4. The legends for Figure 6 should be Llama instead of LAMA

There are inconsistencies between figures. For example,

1. Figure 1 and Figure 4 shows the same evaluation, i.e., the percentage of samples where irrelevant prompts are closer to the edits compared to semantically similar prompts. However, For T5small, Figure 1 shows 46.8% while Figure 4 shows more than 60% in layer 2. Llama has bigger percentage than gpt2 in Figure 1 while has smaller percentage in Figure 4.

**Essential References Not Discussed:**

N/A

**Experimental Designs Or Analyses:**

Please refer to other sections.

In addition, MEMIT as a baseline is not discussed in the paper, e.g., not in the Related Work section. It's not clear why the authors include this one as a baseline.

**Methods And Evaluation Criteria:**

1. For locality, do you compute the train retain rate or the test retain rate (c.f. GRACE paper)?
2. It's not clear how the codebook is used in the experiments. If the stored values are strings as mentioned in the paper, then for edited prompts, there is no need to go through the full autoregressive generation process, but directly output the stored value in the codebook. The authors also mentioned that the stored values can be vectors or LORA indices, how to implement this?
3. It's not clear which token embedding is used for training the projection network. The authors only mentioned that the projection network is applied to only one certain layer, but it's not clear the embedding of which token in the prompts / answers is used for training.
4. Many typos / errors in the methodology section. For example,
+ equation 2, why argmin has underscore value v_i?
+ Threshold in line before equation 2 is v_i^delta, while it's v_delta^i in equation 2.
+ In equation 3, x_i and p_ij are both strings according to definition in Section 3, what is a string vector (with an arrow on top)? how to compute the euclidean distance between two strings?
5. The 4.3 title is misleading. There is no learning process to determine the threshold and tau. tau is a hyperparameter and the threshold depends on the paraphrases chosen.
6. How to choose which layer for training?
7. Have the authors tried other contrastive learning losses, e.g., InfoNCE, etc.?
8. In line 223, why the formulation 3 allows the method to achieve an optimal balance between generalization and locality preservation? How is this optimal? It seems that this formulation can only help generalizability since it preserve the most far away paraphrase which may also include irrelevant edits, thus hurt the locality, which is shown in Table 1 for Counterfact Loc metric.

**Other Comments Or Suggestions:**

Please refine the writing and the presentation of the paper.

**Other Strengths And Weaknesses:**

N/A

**Questions For Authors:**

Please refer to other sections.

**Relation To Broader Scientific Literature:**

Model editing is a trending topic in LLM research. There are many previous works. Many papers have shifted from batch editing to lifelong / continual editing. In this paper, the authors also show results in this setting. Why the Para metric is a bit low on the easier zsRE dataset?

**Theoretical Claims:**

N/A

---

> ### Author Rebuttal · Authors · 2025-04-01
>
> **Comments: There is no definition given for Lexical Bias in the paper.**
>
> **The evidence presentation is a bit messed up. For example,
> Figure 2 uses "Irrelevant Prompts" while Figure 3 uses "Neighbors". ....... should be Llama instead of LAMA**
>
> **There are inconsistencies between figures. For example,
> Figure 1 and Figure 4 shows the same evaluation,... than gpt2 in Figure 1 while has smaller percentage in Figure 4.**
>
> We thank the reviewer for pointing out the typographical error; neighbours and irrelevant prompts refer to the same examples. This will be corrected in the final version. We will improve the images so that the bars are separated, we provide colour correction for Figure 5: (https://imgur.com/a/bFPCBEB).
>
> Regarding figures 1 and 4, the experiments were conducted using different random splits, which accounts for the observed discrepancies in the reported numbers. We will update the figures to ensure consistency in the reported percentages.
>
> We define lexical bias as prompts with similar lexical tokens but different semantics that are closer together in the representation space as compared to a prompt and its respective paraphrases.
>
> **Comment: For locality, do you compute the train retain rate or the test retain rate (c.f. GRACE paper)?**
>
> We compute the locality metric on the test set same as GRACE.
>
> **Comment: It's not clear how the codebook ..... LORA indices, how to implement this?**
>
> We apply the projector once after receiving the user input. From that point, the generation process proceeds normally without intervention—unless the similarity between the input and the nearest codebook key falls below a predefined threshold. In such cases, memory retrieval is triggered. As you have pointed out, the values are stored as strings and they are simply appended to the end of the generated sequence. These values are the edits themselves. Learned vectors can be used to steer the model's generation by adding them to the token representations after identifying the edit scope. Similarly, LoRA blocks can be trained per edit to modulate generation in response to each edit.
>
> **Comment: It's not clear which token embedding is used for training the projection network. The authors only mentioned that the projection network is applied to only one certain layer, but it's not clear the embedding of which token in the prompts / answers is used for training.**
>
> We use averaged token representations for input encoding. While representations of the final token can also be used to train PENME, they tend to be heavily biased, often inflating similarity between prompts that share the same ending token.
>
> **Comments: Many typos / errors in the methodology section. For example,
> equation 2, .........how to compute the euclidean distance between two strings?**
>
> We apologize for the typographical errors. In Equation (2), the argmin operation applies only to the keys $k_i$. The variables $\vec{x_i}$ and $\vec{p_{ij}}$ refer to the output representations from the projection network, which these implied through text rather than a formal definition. We will revise the text to clarify this and eliminate any confusion.
>
> **Comments: The 4.3 title is misleading. There is no learning process to determine the threshold and tau. tau is a hyperparameter and the threshold depends on the paraphrases chosen.**
>
> We will replace "Learning" from the title with   "Finding" to precisely reflect the content of the section.
>
> **Comment: Have the authors tried other contrastive learning losses ...thus hurt the locality, which is shown in Table 1 for Counterfact Loc metric.**
>
> The balance is implied in terms of average score for generalization and locality in comparison with other approaches, where there is a significant deterioration of one metric as compared to the other. We initially evaluated several contrastive learning approaches and found that none consistently outperformed our formulation. This result aligns with the findings of [1], which demonstrate that, in general, the performance of various metric learning methods tends to be comparable.
>
> **Comment: In addition, MEMIT as a baseline ... include this one as a baseline.**
>
> MEMIT is a weight modifying approach which improves over ROME with adjustments to model parameters for all layers instead of one. It is a high performing baseline for weight modifying approaches. We will add the citation alongside ROME for the final version. Currently, the work is cited in experimental setup details provided in Appendix D.1. Experimentation Setup.
>
> **Comment: Why the Para metric is a bit low on the easier zsRE dataset?**
>
> This is for zero-shot transfer from CounterFact to zsRE where some drop in performance is expected. Despite no training on zsRE, PENME still outperforms most methods.
>
> [1]Musgrave, Kevin, Serge Belongie, and Ser-Nam Lim. "A metric learning reality check." In Computer Vision–ECCV 2020: 16th European Conference"

---

> > ### Comment · Reviewer_cuog · 2025-04-06
> >
> > Thank you for your rebuttal.
> >
> > Some of my concerns are still there, please see below:
> >
> > 1. I agree with reviewer emXV on the usefulness of PENME in real knowledge editing setting, especially in the sequential editing setting.  Since the error is not addressed within the model itself, we need to apply this key searching step manually. As suggested by the authors, this is applied every time when the model receives a user prompt. The authors also showed in the rebuttal to emXV that this edited info could be retained by using a specific prompt. However, it would be difficult In the sequential editing scenario. If we first edit the born city of person P1 from C1 to C2, then edit the director of a certain institute I1 from P2 to P1, and query the model which city was the director of I1 born. It would be hard to retrieve all the related edits and put them into the prompt in the current pipeline.
> >
> > 2. My question about the optimality is more about the equation itself, not about the effectiveness of the empirical results compared to other baselines. To be more specific, why equation 3 is optimal compared to the equation in line 226? or what if we don't use max in equation 3 but use median / mean in equation 3? or what if we use both information from semantic relevant and irrelevant prompts to get a balance?
> >
> > Please feel free to share your thoughts!

---

> > > ### Author Response · Authors · 2025-04-06
> > >
> > > **Comment: I agree with reviewer emXV on the usefulness of PENME in real knowledge editing setting, especially in the sequential editing setting. Since the error is not addressed within the model itself, we need to apply this key searching step manually. As suggested by the authors, this is applied every time when the model receives a user prompt. The authors also showed in the rebuttal to emXV that this edited info could be retained by using a specific prompt. However, it would be difficult In the sequential editing scenario. If we first edit the born city of person P1 from C1 to C2, then edit the director of a certain institute I1 from P2 to P1, and query the model which city was the director of I1 born. It would be hard to retrieve all the related edits and put them into the prompt in the current pipeline.**
> > >
> > > Thank you for the discussion. We would like to emphasize that model editing remains a fundamentally challenging problem, particularly due to the difficulty of modifying pre-trained models without compromising their overall performance. The scenario highlighted by the reviewer involves multi-hop edits—cases where the model must accommodate multiple, interdependent changes. We highlight that such edits, even when applied locally within the model’s parameters (weight modification), do not propagate broadly across the network. The best performance achieved in this context is around 5\% by AlphaEdit [1]. However, this statistic can be misleading, as shown by IKE [2], which demonstrates that this performance, when viewed in a broader context, actually represents a degradation of the model's original multihop reasoning capabilities. This is consistent with findings by [3], which demonstrate that even localized weight updates
> > > made for editing leads to both gradual and catastrophic forgetting.
> > >
> > > In light of this, PENME provides a promising direction by focusing on weight-preserving techniques, aiming to mitigate drastic performance loss while enhancing the model's ability to handle sequential edits. As mentioned in our response to reviewer Qzbk, in scenarios where edits are related and require combined information for generation, these edits can be linked together in PENME's codebook for efficient retrieval. Using this with ICL will allow for the model to reason using the complete information within the prompt. We agree with the reviewer that integrating multiple edits into a single prompt can be problematic, especially for smaller models with limited input sequence lengths. We humbly mention that all current model editing approaches involve inherent compromises, and we do not claim to fully solve the problem of model editing but instead present a concrete improvement over weight-preserving methods while also highlighting the key issue as to why they fail. We appreciate the discussion on multi-hop performance and will add it to the paper.
> > >
> > > [1] Fang, Junfeng, Houcheng Jiang, Kun Wang, Yunshan Ma, Xiang Wang, Xiangnan He, and Tat-seng Chua. "Alphaedit: Null-space constrained knowledge editing for language models."
> > >
> > > [2] Zheng, Ce, Lei Li, Qingxiu Dong, Yuxuan Fan, Zhiyong Wu, Jingjing Xu, and Baobao Chang.
> > > ”Can We Edit Factual Knowledge by In-Context Learning?.”
> > >
> > > [3] Gupta, Akshat, Anurag Rao, and Gopala Anumanchipalli. "Model Editing at Scale leads to Gradual and Catastrophic Forgetting."
> > >
> > > **Comment: My question about the optimality is more about the equation itself, not about the effectiveness of the empirical results compared to other baselines. To be more specific, why equation 3 is optimal compared to the equation in line 226? or what if we don't use max in equation 3 but use median / mean in equation 3? or what if we use both information from semantic relevant and irrelevant prompts to get a balance?**
> > >
> > > When incorporating both relevant and irrelevant prompts using the mean or median for Equations 3 and in line 226, the threshold can be logically set to the midpoint. While this approach may seem to offer a better balance, the distances between edits and irrelevant prompts vary after training, resulting in a non-uniform distribution. As a result, some similarity thresholds become large for certain edits, which increases the risk of locality failures. We explored this approach in the early stages of the work, but it did not yield the desired outcomes, leading us to refine our method for improved performance. We will add empirical results comparing the results for these equations in the paper.

---

### Official Review · Reviewer_jhu1 · 2025-03-13

**Overall Recommendation:** 3

**Summary:**

This paper proposes PENME a learnable projection network to transform the model’s internal representation such that lexical bias during model tuning is minimized. In particular, the authors found that existing adapters often misfire on inputs that share words with a stored edit but aren’t actually the same fact, while also failing to generalize to paraphrases. To address this PENME attempts to minimized & pull semantic paraphrases closer while lexically similar prompts are pushed farther apart. Additionally, PENME shows strong empirical results on several benchmarks with significantly better generalization abilities and computationally efficiency.

**Claims And Evidence:**

The paper's claims are clear with some experimental results to backup the idea of PENME having general improvements over prior methods.

**Essential References Not Discussed:**

The reviewer is unaware of other necessary references which are not linked.

**Experimental Designs Or Analyses:**

The chosen benchmarks are fair and follow in line with prior works. Additionally, the set of comparative methods is broad alongside a range of model architectures. Finally there are also a large amount of ablation studies which cover a series of supporting analysis.

**Methods And Evaluation Criteria:**

PENME as a solution to the given problem is well motivated and the idea of inserting a learned projection module to re-map the latent space for editing is intuitive. Additionally, the chosen counterfactual and zsRE datasets are standard benchmarks from prior works. Overall, the reviewer has no high level concern with the proposed methodology and experimental design.

**Other Comments Or Suggestions:**

See the sections above.

**Other Strengths And Weaknesses:**

The paper establishes the issue of lexical bias in model editing which provides strong insights. Additionally the PENME framework is a intuitive next step. However, this leads PENME's contributions to be less significant methodologically.

**Questions For Authors:**

No additional questions beyond the issues raised in the sections above.

**Relation To Broader Scientific Literature:**

The paper positions itself well in the context of prior model editing research. The observations of lexical bias is also an under explored area of research and would help broaden the scope of the model editing.

**Theoretical Claims:**

The paper has no direct theoretical claims and therefore rigorous proofs are not relevant.

---

> ### Author Rebuttal · Authors · 2025-04-01
>
> **Comment: The paper establishes the issue of lexical bias in model editing which provides strong insights. Additionally the PENME framework is a intuitive next step. However, this leads PENME's contributions to be less significant methodologically.**
>
> While we understand that PENME may appear as a natural extension, we respectfully argue that our work addresses a fundamental and previously underexplored challenge in the weight-preserving model editing paradigm—namely, the problem of scoping, i.e., determining when an input should invoke the edited knowledge. Our proposed scoping mechanism contributes to the weight-preserving model editing paradigm and is not limited to our editing method, PENME. Moreover, the lexical bias has been highlighted as a substantial issue in text and vision models in the recent paper [1]. Our work is the first work that attempts to address this issue.
>
> [1] Dumpala, S. H., Jaiswal, A., Sastry, C., Milios, E., Oore,
> S., and Sajjad, H. SUGARCREPE++ Dataset: Vision Language Model Sensitivity to Semantic and Lexical Alterations. In Conference on Neural Information Processing Systems, Dataset Track (NeurIPS), 2024.

---

### Official Review · Reviewer_Qzbk · 2025-03-14

**Overall Recommendation:** 4

**Summary:**

This paper addresses the challenge of lexical bias in adapter based model editing for large language models LLMs Specifically the authors identify an important limitation in current methods where irrelevant prompts those lexically similar but semantically unrelated to edited prompts are prone to misfires negatively impacting edit locality.
To mitigate this the authors propose PENME Projector Editor Networks for Model Editing a framework that introduces a contrastively trained projection network to better disentangle prompt representations.
By learning to group semantically similar prompts while distancing lexically similar but unrelated ones PENME improves both locality and generalization in model editing tasks.
The method integrates into existing adapter based key value codebooks preserving the efficiency and modularity of these approaches while offering notable improvements in edit precision Experiments on standard benchmarks zsRE and Counterfact across three prominent model architectures T5-small GPT2-XL and Llama-2-7B demonstrate that PENME consistently outperforms state of the art methods such as GRACE and MELO Additionally the authors discuss scalability and codebook management which have been concerns in prior approaches.

## update after rebuttal
This paper proposed a method address the lexical bias in model editing task. The response mostly address my concerns, and I keep my original rating.

**Claims And Evidence:**

The paper makes the following key claims:

1, Lexical dominance exists in existing adapter based editing systems and leads to locality degradation

2, PENME reduces lexical bias thereby improving both locality and paraphrase generalization

3, PENME outperforms prior methods in standard evaluation metrics on zsRE and Counterfact datasets

4, The framework is scalable requiring fewer codebook entries and reduced management overhead

In my assessment these claims are largely substantiated
* The authors present convincing empirical evidence particularly the layer wise lexical dominance analysis Figure 4 which demonstrates the motivation behind their approach
* Quantitative results in Table 1 reflect consistent improvements across multiple evaluation metrics
* The ablation studies and scalability analyses Figures 5 and 6 Section 4 offer further support for the systems design decisions
* The evidence is thorough and the experimental results align well with the claims made

**Essential References Not Discussed:**

N.A.

**Experimental Designs Or Analyses:**

The experimental design is robust
* Multiple LLM architectures are considered increasing the generality of the findings
* The use of both batch editing and streaming edit scenarios demonstrates flexibility
* The ablation studies on key hyperparameters τ ϕ are comprehensive and provide insight into the tradeoffs inherent in the approach
* Results are presented clearly with appropriate baselines GRACE MELO SERAC MEMIT included for comparison
My suggestion here would be to extend the evaluation to additional datasets perhaps outside of fact based knowledge editing to explore how well PENME generalizes.

**Methods And Evaluation Criteria:**

The methodology is appropriate and thoughtfully designed Leveraging contrastive learning to train the projection network is a natural choice for encouraging representation disentanglement. The projection network is lightweight a simple two layer MLP which ensures efficiency

The evaluation metrics edit success locality and paraphrase generalization are standard in model editing research and appropriate for this work. The datasets selected zsRE Counterfact are commonly used and offer a reasonable degree of diversity in task difficulty.

**Other Comments Or Suggestions:**

* Figures 1 and 4 could be made clearer axis labels and figure legends are somewhat difficult to read
* The authors may wish to include qualitative examples of edits for example paraphrases handled correctly to better illustrate practical gains
* There is an opportunity to discuss potential defensive mechanisms to mitigate malicious editing

**Other Strengths And Weaknesses:**

Strengths:
- The problem of lexical bias in model editing is underexplored and the authors offer a practical well motivated solution
PENME demonstrates clear empirical improvements over strong baselines
- The method scales well and reduces codebook management complexity which addresses a common criticism of adapter based approaches
 - The paper is well written and accessible even for readers not deeply familiar with model editing.

Weaknesses:
- The contribution is incremental rather than fundamental it represents an important refinement to adapter based editing but does not propose a fundamentally new editing mechanism
- The applicability to complex generation tasks for example long form generation reasoning tasks is not explored

**Questions For Authors:**

Q1, How does PENME perform on long form generation tasks where maintaining locality over extended sequences may be challenging?

Q2, Could the authors comment on PENMEs generalization under domain shift for example specialized domains like healthcare or legal texts?

Q3, Have the authors considered adversarial training to further enhance the robustness of locality and paraphrase generalization?

Q4, How might PENMEs projection space handle multi hop reasoning prompts where lexical and semantic relationships are more complex?

**Relation To Broader Scientific Literature:**

The paper builds directly on prior adapter based editing methods GRACE, MELO while drawing from contrastive learning principles common in representation learning.

**Theoretical Claims:**

This paper does not make formal theoretical claims that require rigorous proof verification. However the conceptual grounding of PENME is sound and well supported by empirical evidence.

---

> ### Author Rebuttal · Authors · 2025-04-01
>
> **Comments: The contribution is incremental rather than fundamental it represents an important refinement to adapter based editing but does not propose a fundamentally new editing mechanism**
>
> **The applicability to complex generation tasks for example long form generation reasoning tasks is not explored**
>
> **Q1, How does PENME perform on long form generation tasks where maintaining locality over extended sequences may be challenging?**
>
> We appreciate the reviewer’s perspective. However, we respectfully argue that our work addresses a fundamental and previously underexplored challenge in the weight-preserving model editing paradigm—namely, the problem of scoping, i.e., determining when an input should invoke the edited knowledge. Our proposed scoping mechanism contributes to the weight-preserving model editing paradigm and is not limited to our editing method, PENME. Moreover, the lexical bias has been highlighted as a substantial issue in text and vision models in the recent paper [1]. Our work is the first work that attempts to address this issue.
>
> We did not test PENME on long-form generation due to limited available datasets and baselines for meaningful comparison. Long-form generation may require incorporating techniques such as LORA or other lightweight trainable components. We believe that this is out of the scope of the current focus of the paper and is a promising direction for future work.
>
> **Comment: Figure 3 and Figure 7 illustrate the "Percentage of samples where edits are closer to unrelated neighbors," but this is insufficient to demonstrate lexical bias. At lower model layers, high similarity may result from underdeveloped sentence representations, while at higher layers, the reduced percentage indicates greater differentiation between sentences.**
>
>
> We agree with the reviewer that lower layers of a model may encode less developed sentence representations, while higher layers tend to capture more abstract semantic similarity. However, as shown in the bar chart in Figure 3, lexical bias persists even in the higher layers, suggesting that this issue is not confined to shallow representations. Furthermore, similar findings have been reported by [1], who observed significant lexical bias in representations across a diverse range of vision and text language models.
> These observations reinforce the broader relevance of the issue we address in this work.
>
> **Comment: Q2, Could the authors comment on PENMEs generalization under domain shift for example specialized domains like healthcare or legal texts?**
>
> We have evaluated PENME's generalization capabilities by assessing performance transfer from the CounterFact to the zsRE dataset. While our current experiments focus on general QA data, the results suggest that PENME can adapt to new domains with minimal supervision. We hypothesize that for specialized domains such as healthcare or legal texts, with a small amount of in-domain data, PENME would generalize effectively to any domain.
>
> **Comment: Q3, Have the authors considered adversarial training to further enhance the robustness of locality and paraphrase generalization?**
>
> The CounterFact dataset used in our experiments is inherently adversarial, as irrelevant prompts often share significant lexical and semantic overlap (e.g. the subjects are both actors) with the edited facts. Additional adversarial examples could indeed be constructed using larger language models to further enhance robustness and generalization. However, generating such data at scale would increase computational cost and resource requirements for training. This is a valuable future direction that leads the way towards PENME generalization to a wide set of domains.
>
> **Comment: Q4, How might PENMEs projection space handle multi hop reasoning prompts where lexical and semantic relationships are more complex?**
>
> We posit that a knowledge base will be needed, and such prompts would need to be stored in the codebook with links to the main edit. PENME's learned representation space would allow it to localize this information better in the representation space, thus allowing for scoping multi-hop or one-hop questions. At inference time, these can be used to link back to the edited information, which can then be used for informed generation through some training of additional parameters, e.g. LoRA block.
>
>
> [1] Dumpala, S. H., Jaiswal, A., Sastry, C., Milios, E., Oore,
> S., and Sajjad, H. SUGARCREPE++ Dataset: Vision Language Model Sensitivity to Semantic and Lexical Alterations. In Conference on Neural Information Processing Systems, Dataset Track (NeurIPS), 2024.

---

### Official Review · Reviewer_emXV · 2025-03-23

**Overall Recommendation:** 2

**Summary:**

This paper is about parameter preserving knowledge editing methods, specifically adapter based methods for knowledge editing. They show that adapters have lexical bias, which is the vulnerability to recalling unrelated facts due to overlapping n-grams. To mitigate this, it proposes Projector Editor Networks for Model Editing (PENME), which employs contrastive learning to learn a disentangled representation space, effectively separating lexical and semantic similarities.

**Claims And Evidence:**

Claim 1 :  There is lexical bias in adapter recall
Evidence : shown through n-gram overlap and Figure 4


Claim 2: Use contrastive learning and projector network to disentangle representations of unrelated facts
Evidence : Shown through editing performance in Table 1. But the improvement is not as clear for CounterFact. Can the authors explain that in more detail?

**Essential References Not Discussed:**

Consecutive Model Editing with Batch alongside HooK Layers - Very related method that should have been one of the baselines which came out in March 2024.

**Experimental Designs Or Analyses:**

Experimental design was standard. The only issue was that no downstream performance analysis done.

A question for the authors here is - How do these edits interfere with other task performance and other scenarios like general text generation task? Is the projection mechanism activated at the generation of every token?

**Methods And Evaluation Criteria:**

They use batch and sequential editing using zsre and counterfact dataset. This is standard in knowledge editing.

**Other Comments Or Suggestions:**

NA

**Other Strengths And Weaknesses:**

NA

**Questions For Authors:**

I have the following questions for the authors:

    1. How are the values incorporated since they are stored as string? No presentation of how inference is done? What is the inference process?
    2. How does the key storage process work for other methods? How is it different? I dont really know GRACE or MELO?
    3. How does this work with general generation?  Is the projection mechanism activated at the generation of every token?

I am happy to update my score based on the author's responses to the above questions and other questions I asked throughout the review.

**Relation To Broader Scientific Literature:**

This paper does present a viable addition to adapter-based knowledge editing methods.

The reason why parameter preserving methods are attractive is because parameter modifying methods lead to model degradation. But recent knowledge editing methods have made impressive strides in this, preserving downstream performance over 3k sequential edits (AlphaEdit). Performing evaluations for 2k edits is does no longer enough and a good measure of effectiveness for these methods.

So to show significant addition to the literature would be showing the effectiveness of this method on a much larger set of edits and also measuring downstream performance. Discussing how these methods interact with downstream performance is also important for that.

**Theoretical Claims:**

Mostly an empirical study

---

> ### Author Rebuttal · Authors · 2025-04-01
>
> **Comment:"The only issue was that no downstream performance analysis done."**
>
> We evaluate the general capabilities of the Llama-2-7b used in our study and compare its downstream performance before and after applying PENME. To ensure a diverse and representative assessment, we select three distinct tasks: Natural Language Inference-NLI (RTE dataset) evaluated using F1 score, Summarization (CNN/DailyMail) assessed via average ROUGE-1, ROUGE-2, and ROUGE-L scores, and Sentiment Classification (Diar-AI/Emotions),  using F1 score.
>
> | **Task**                        | **Baseline** | **Edited Model** |
> |--------------------------------|--------------|------------------|
> | NLI | 0.6476       | 0.6428           |
> | Text Classification            | 0.6573       | 0.6573           |
> | Summarization                  | 0.1865       | 0.1865           |
>
> The results show competitive performance across all tasks, with only a small drop in performance
> in the NLI task.
>
> **Comments:" How do these edits interfere... generation of every token?**
>
>  **In discussing the construction ... didn't explain how to obtain the corresponding values.**
>
> **How are the values incorporated... What is the inference process?**
>
> **How does this work .... generation of every token?"**
>
> We apply the projector once after receiving the user input. From that point, the generation process proceeds normally without intervention—unless the similarity between the input and the nearest codebook key falls below a predefined threshold. In such cases, memory retrieval is triggered. As you have pointed out, the values are stored as strings and they are simply appended to the end of the generated sequence. These values are the edits themselves. We did not evaluate generative settings where multi-token or continued generation was necessary. Instead of using a stored string, we could use the approaches examined in GRACE, where a learned vector is the value, and MELO, where a LoRA block is the value.  As a point of emphasis, our proposed scoping mechanism is independent of these design choices, and any generation method can be freely selected.
>
> **Comment:"The reason why parameter preserving .... evaluations for 2k edits is does no longer enough .... effectiveness for these methods."**
>
> While AlphaEdit does improve upon MEMIT, it is important to clarify that the “sequential”
> setting used in AlphaEdit involves 3,000 edits applied in batches of 100, making it more accurately
> described as consecutive batch editing. In contrast, true sequential editing—where updates are applied
> one at a time, as explored in GRACE—poses a more challenging scenario due to the cumulative nature
> of parameter shifts (3000 vs 30 updates). We provide results for 3000 sequential edits on the Llama-2-7b model. PENME shows high performance with a minor drop in generalization and locality metrics.
> | **ES** | **Para** | **Loc** |
> |------------------|---------------------------|--------------|
> | 1.00             | 0.8076                    | 0.8549       |
>
> **Comment: "Consecutive Model Editing ....one of the baselines which came out in March 2024"**
>
> We thank the reviewer for highlighting this work. While the initial codebase of the work supports
> GPT series models, adapting it to LLaMA requires additional effort. Notably, Table 1 in the paper shows
> lower generalization and edit success performance on GPT2-XL compared with our method, PENME.
> We will add the comparison in the final version of the paper.
>
> **Comments:"How does the key storage ... I dont really know GRACE or MELO?**
>
> **Claim 2: Use contrastive learning .... But the improvement is...**
> **not as clear for CounterFact. ..."**
>
> GRACE and MELO store direct model representations corresponding to the final hidden token of the input. Their approach relies on maintaining a very low similarity threshold, which necessitates adding a large number of entries to the codebook per edit. To mitigate this overhead, they merge entries with similar outputs. However, when new edits fall within the similarity threshold, the similarity thresholds are reduced for these edits. This threshold adjustment and merging strategy can result in edit forgetting, as discussed in the reference section.
>
> In contrast, PENME avoids the need to store multiple codebook entries by learning a projection space where irrelevant inputs are mapped farther from each other, while paraphrased edits remain close. This design enables faster retrieval and helps prevent issues like edit clashes and forgetting, without compromising precision.
>
> We evaluate GRACE by increasing its similarity threshold on GPT2-XL, aiming to match the generalization performance of PENME. The results overall demonstrate how PENME improves upon the direct use of representations.
>
> | Model | ES   | Loc   | Para  |
> |-------|------|-------|-------|
> | PENME | **1.00** | **0.847** | **0.875** |
> | GRACE | **1.00** | 0.171 | 0.767 |

---

> > ### Comment · Reviewer_emXV · 2025-04-01
> >
> > Thank you for your responses.
> >
> > I'm still concerned about the usefulness of PENME. The rebuttal and the paper suggests to me that PENME is a very specific method created to be great at the **metrics** of the task of knowledge editing, but it does not solve the problem of knowledge editing. My main criticism/skepticism is due to two things:
> >
> > 1. **Rebuttal Quote -"The values are stored as strings and they are simply appended to the end of the generated sequence"**
> >
> > What happens if I want the model to generate a paragraph based on the edited facts? If the value is just appended as a string to the input, it may or may not be grammatically correct or an appropriate continuation of generation. Whereas if the values were hidden activations used to guide the model, then the model's generation uses it for text generation. I think this is completely missed in PENME unless I understand it incorrectly. This property itself will make PENME great at knowledge editing evaluation where inputs are Question-Answer pairs, but is not useful when using an LLM.
> >
> >
> > 2. **Rebuttal Quote - "We apply the projector once after receiving the user input. From that point, the generation process proceeds normally without intervention"**
> >
> > What if during the generation process, the model reaches a point where it needs to continue generation based on an edited fact. With chain of thought and test-time scaling, such scenarios will occur at a much larger frequency. If the memory retrieval is only triggered once when the input is given to the model, such a scenario is completely ignored.
> >
> > Please feel free to provide more clarification on the above.

---

> > > ### Author Response · Authors · 2025-04-02
> > >
> > > You are right that using a string as a value can lead to issues with grammatical correctness or appropriate continuation. However, the datasets used for model editing are question-answer based, where the naturally generated answer is typically a single token or phrase. As we mentioned in the paper, alternative methods such as LoRA (MELO), vector playback (GRACE), can be integrated seamlessly in PENME's values, which allow for scenarios where the model can continue generation based on an edited fact. We didn't evaluate this approach in practice, but we will add the results to the camera ready paper if accepted, as it is more work than what can be completed during the rebuttal period.
> > >
> > > We emphasize that our **primary contribution** lies in addressing a fundamental and critical point of failure in weight-modified approaches—namely, the scoping mechanism. Our work tackles this core and previously underexplored challenge within the weight-preserving model editing paradigm. Importantly, the accuracy of any subsequent generation process is inherently dependent on resolving this issue.
> > >
> > > **To address the reviewer’s concern** we propose an alternative generation mechanism that integrates in-context learning within PENME framework to support long-form generation. For this setup, PENME stores an edit prompt as the value in its codebook. Since PENME is integrated into the early layers of the model, computation can be terminated early if the input falls within the scope of an edit, allowing the relevant edit information to be efficiently retrieved from the codebook. Once obtained, this edited information can be incorporated into the prompt context to support in context learning (ICL) based generation, as demonstrated by [1][2], who validated its effectiveness for long-form generation.
> > >
> > > We apply the proposed method using LLaMA-2-7B model considering 1,000 samples from the Counterfact dataset. We follow the standard evaluation settings described in [3] i.e. maximum token length is set to $100$ and assessed Fluency (via Generation Entropy) and Consistency (via Reference Score) of the generated output in comparison to the baseline model's output. In the following, we also provide sampled generated outputs with max token length set to $300$.
> > >
> > > We use the following prompt, which is stored as the value in the codebook:
> > > ```
> > > f"""You have a new fact: \{edit prompt\}.
> > > Based on this fact, complete the following sentence to answer the question: \{query\}
> > > Your answer should specifically incorporate the new fact I've shared.
> > >
> > > Paragraph: \{query\}"""
> > > ```
> > >
> > > | Prompt Edit                                     | Original Information | Edit    | Query                                           | Answer |
> > > |-|-|-|-|-|
> > > | What is the twin city of Shanghai? It is       | Barcelona | Dresden | The twin city of Shanghai is none other than     | The twin city of Shanghai is none other than **Dresden** Germany. This may come as a surprise to many as **Dresden** is located in Eastern Germany while Shanghai is a major city in Eastern China. However, the two cities have a long history of cultural and economic ties dating back to the 19th century when both were part of the German Empire. In the years following World War II, **Dresden** and Shanghai maintained close diplomatic relations, and in 1981 the two cities officially designated each other as twin cities. Today, the connection between **Dresden** and Shanghai continues to flourish with regular exchanges in the fields of education, culture, and trade. |
> > > | Baal Shem of London speaks the language         | Hebrew| French  | Baal Shem of London is proficient in             | Baal Shem of London is proficient in multiple languages including **French**. This is a remarkable feat given that he is a spiritual leader and healer who is said to have the ability to communicate with the divine. However his proficiency in **French** is particularly noteworthy as it is a language that is not commonly associated with his spiritual tradition. Despite this Baal Shem of London has demonstrated a remarkable ability to master this language allowing him to connect with and heal individuals from a diverse range of cultural backgrounds. This is yet another testament to his incredible spiritual abilities and his commitment to serving others. |
> > >
> > > The generated text shows that the edited fact is propagated throughout the generation, and the generation is coherent.
> > >
> > > | Model    | Generation Entropy | Consistency |
> > > |-|-|-|
> > > | Baseline | 611.54              | 16.57       |
> > > | Edited   | 622.36              | 21.98       |
> > >
> > > The Generation Entropy increases from 611.54 to 622.36, indicating more diverse and fluent text. The Consistency score also improves from 16.57 to 21.98, which means generations are more semantically consistent.
> > >
> > > [1] Rosati, D. et al. Long-form evaluation of model editing
> > >
> > > [2] Zheng, Ce et al. Can We Edit Factual Knowledge by In-Context Learning?.
> > >
> > > [3] Meng, K. et al. Mass-editing memory in a transformer.

---

### Decision · Program_Chairs · 2025-05-01

**Decision:**

Accept (poster)

**Comment:**

This paper introduces a new approach for model editing, focusing on a new scoping mechanism for weight-preserving editors to decide when edits should be applied. This helps mitigate the flaw they identify with prior works: Edits are often falsely triggered for inputs that have n-grams that are present in prior edits, even when they are semantically distinct. Their approach learns to project edits into a new space in which their semantic differences are emphasized, leading to better localization while maintaining generalization of the edits.

Reviewers overall lean positively, though there are some concerns. On the positive side, reviewers appreciated the novelty of the solution, the identified weakness of prior works, sufficient experimental comparisons, and the outperformance of alternatives. On the negative side, reviewers are concerned by a lack of downstream evaluations (which is a big, recent focus of model editing works) and that the practical implications may be limited since model edits are only for QA tasks. They also note some opportunities to further clarify the writing (e.g., "lexical bias" is in the title, but isn't clearly defined in the paper).

Overall, the experiments appear sound and that advances beyond the state-of-the-art are compelling, so I believe the strengths outweigh the weaknesses and suggest acceptance.